



# The 852/3 CE Mount Churchill eruption: examining the potential climatic and societal impacts and the timing of the Medieval Climate Anomaly in the North Atlantic Region

Helen Mackay[1], Gill Plunkett[2], Britta J.L. Jensen[3], Thomas J. Aubry[4], Christophe Corona[5,6], Woon Mi Kim[7,8], Matthew Toohey[9], Michael Sigl[7,8], Markus Stoffel[6,10,11], Kevin J. Anchukaitis[12], Christoph Raible[7,8], Matthew S. M. Bolton[3], Joseph G. Manning[13], Timothy P. Newfield[14,15], Nicola Di Cosmo[16], Francis Ludlow[17], Conor Kostick[17], Zhen Yang[17], Lisa Coyle McClung[2], Matthew Amesbury[18], Alistair Monteath[3], Paul D.M. Hughes[19], Pete G. Langdon[19], Dan Charman[18], Robert Booth[20], Kimberley L. Davies[21], Antony Blundell[22], and Graeme T. Swindles[23]

[1]Department of Geography, Durham University, Durham, DH1 3LE, UK
[2]Archaeology & Palaeoecology, School of Natural and Built Environment, Queen's University Belfast, Belfast, BT7 1NN, UK
[3]Earth and Atmospheric Sciences, Faculty of Science, University of Alberta, Edmonton, Alberta, T6G 2E3, Canada
[4]Department of Geography, University of Cambridge, UK; Sidney Sussex College, Cambridge, CB2 3HU, UK
[5]Université Clermont-Auvergne, Geolab UMR 6042 CNRS, France
[6]Institute for Environmental Sciences, University of Geneva, CH-1205 Geneva, Switzerland
[7]Climate and Environmental Physics, Physics Institute, University of Bern, Bern, CH-3012, Switzerland
[8]Oeschger Centre for Climate Change Research, University of Bern, Bern, CH-3012, Switzerland
[9]Institute of Space and Atmospheric Studies, University of Saskatchewan, Saskatoon, S7N 5E2, Canada
[10]Department of Earth Sciences, University of Geneva, CH-1205 Geneva, Switzerland
[11]Department F.-A. Forel for Environmental and Aquatic Sciences, University of Geneva, CH-1205 Geneva, Switzerland
[12]School of Geography, Development, and Environment and Laboratory of Tree-Ring Research, University of Arizona, Tucson, AZ 85721 USA
[13]Department of History, Yale University, New Haven, 06520, USA
[14]Department of History, Georgetown University, Washington, 20057, USA (TN)
[15]Department of Biology, Georgetown University, Washington, 20057, USA (TN)
[16]Institute for Advanced Study, Princeton, New Jersey, 08540, USA
[17]Department of History, and Trinity Centre for Environmental Humanities, Trinity College Dublin, Dublin, D02 PN40, Ireland.
[18]Geography, College of Life and Environmental Sciences, University of Exeter, EX4 4ST, UK
[19]School of Geography and Environmental Science, University of Southampton, SO17 1BJ, UK
[20]Earth and Environmental Science Department, Lehigh University, Pennsylvania, 18015, USA
[21]Institute for Modelling Socio-Environmental Transitions, Bournemouth University, Bournemouth, BH12 5BB, UK
[22]School of Geography, University of Leeds, Leeds, LS2 9JT, UK
[23]Geography, School of Natural and Built Environment, Queen's University Belfast, Belfast, BT7 1NN, UK

*Correspondence to*: Helen Mackay (helen.mackay@durham.ac.uk)



**Abstract**
The 852/3 CE eruption of Mount Churchill, Alaska, was one of the largest first millennium volcanic events, with a
magnitude of 6.7 (VEI 6) and a tephra volume of 39.4–61.9 km$^3$ (95% confidence). The spatial extent of the ash
fallout from this event is considerable and the cryptotephra (White River Ash east; WRAe) extends as far as Finland
and Poland. Proximal ecosystem and societal disturbances have been linked with this eruption; however, wider
eruption impacts on climate and society are unknown. Greenland ice-core records show that the eruption occurred in
winter 852/3 ± 1 CE and that the eruption is associated with a relatively moderate sulfate aerosol loading, but large
abundances of volcanic ash and chlorine. Here we assess the potential broader impact of this eruption using
palaeoenvironmental reconstructions, historical records and climate model simulations. We also use the fortuitous
timing of the 852/3 CE Churchill eruption and its extensively widespread tephra deposition of the White River Ash
(east) (WRAe) to examine the climatic expression of the warm Medieval Climate Anomaly period (MCA; ca. 950–
1250 CE) from precisely linked peatlands in the North Atlantic region.
The reconstructed climate forcing potential of 852/3 CE Churchill eruption is moderate compared with the eruption
magnitude, but tree-ring-inferred temperatures report a significant atmospheric cooling of 0.8 °C in summer 853 CE.
Modelled climate scenarios also show a cooling in 853 CE, although the average magnitude of cooling is smaller
(0.3 °C). The simulated spatial patterns of cooling are generally similar to those generated using the tree-ring-
inferred temperature reconstructions. Tree-ring inferred cooling begins prior to the date of the eruption suggesting
that natural internal climate variability may have increased the climate system's susceptibility to further cooling.
The magnitude of the reconstructed cooling could also suggest that the climate forcing potential of this eruption may
be underestimated, thereby highlighting the need for greater insight into, and consideration of, the role of halogens
and volcanic ash when estimating eruption climate forcing potential.
Precise comparisons of palaeoenvironmental records from peatlands across North America and Europe, facilitated
by the presence of the WRAe isochron, reveal no consistent MCA signal. These findings contribute to the growing
body of evidence that characterizes the MCA hydroclimate as time-transgressive and heterogeneous, rather than a
well-defined climatic period. The presence of the WRAe isochron also demonstrates that no long-term
(multidecadal) climatic or societal impacts from the 852/3 CE Churchill eruption were identified beyond areas
proximal to the eruption. Historical evidence in Europe for subsistence crises demonstrate a degree of temporal
correspondence on interannual timescales, but similar events were reported outside of the eruption period and were
common in the 9$^{\text{th}}$ century. The 852/3 CE Churchill eruption exemplifies the difficulties of identifying and
confirming volcanic impacts for a single eruption, even when it is precisely dated.



## 1. Introduction

The 852/3 CE eruption of Mount Churchill in the Wrangell volcanic field, southeast Alaska, was one of the largest first millennium volcanic events, with a roughly estimated eruptive volume of 47 km$^3$ and top plume height of ca. 40–45 km (Lerbekmo, 2008). The considerable ash fall-out from this Volcanic Explosivity Index (VEI) 6 Plinian eruption extended eastwards: visible horizons of the ash, termed White River Ash east (WRAe), have been identified >1300 km from the source (e.g. Lerbekmo, 2008; Patterson et al., 2017) and WRAe cryptotephra (non-visible volcanic ash) deposits have been detected in northeastern North America (Pyne O'Donnell et al., 2012; Mackay et al., 2016; Jensen et al., in press; Figure 1a-c). Furthermore, the correlation of the WRAe with the "AD 860B" tephra first identified in Ireland (Pilcher et al. 1996) has extended the known spatial distribution of the cryptotephra to Greenland (NGRIP and NEEM ice cores) and western and eastern Europe (e.g., Coulter et al., 2012; Jensen et al., 2014; Watson et al., 2017a, b; Kinder et al., 2020).

The ash produced from this eruption caused considerable and long-lasting environmental disturbances in regions proximal to Mount Churchill. For example, the eruption has been linked with changes in vegetation that persisted for ca. 50-150 years in Yukon (Rainville, 2016), multi-centennial changes in peatland ecology in southeast Alaska (Payne and Blackford, 2008) and decreases in aquatic productivity lasting ca. 100 years in southwest Yukon (Bunbury and Gajewski, 2013). These spatial patterns in proximal environmental responses to the 852/3 CE Churchill eruption are diverse. The eruption and its environmental impacts are also suggested to have driven societal changes in the region (Kristensen et al., 2020), notably a decline in indigenous occupancy in the southern Yukon (Hare et al., 2004). In addition, the event may have triggered the southwards migration of people, who brought their culture and Athapaskan language to the US Great Basin and the American Southwest (Mullen, 2012). Several studies have therefore characterized the proximal impacts of this 852/3 CE Churchill eruption, but less is known about the widescale Northern Hemisphere (NH) or global impacts of this large eruption.





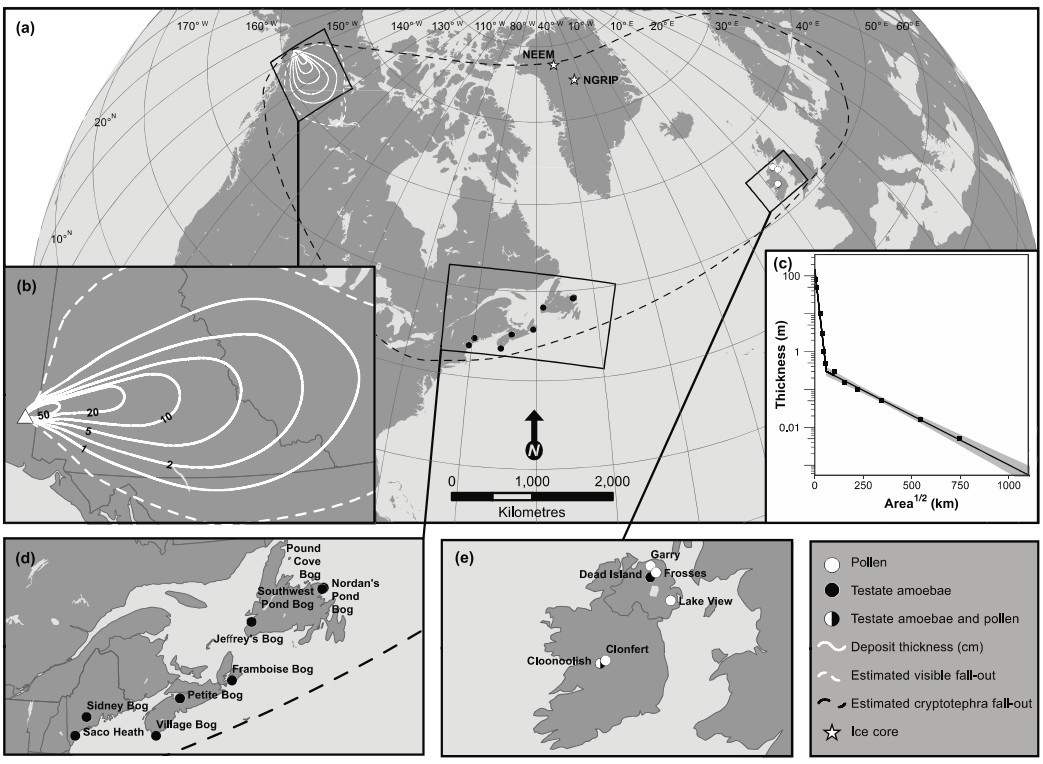

93

**Fig. 1: Site and White River Ash east distribution map with thickness data for volume estimate. (a) Location map, highlighting Greenland ice core sites (NEEM = North Eemian, NGRIP = North Greenland Ice Core Project), estimated cryptotephra fall-out area, and inset map extents. (b) Isopach map synthesized from distal and proximal isopachs (Lerbekmo 1975, 2008). (c) Plot of deposit thickness (on a log scale) against square root area of isopachs ≥ 0.5 cm and two-piece exponential fit (black line). The grey shaded area represents the 95% confidence interval of the fitted function. (d) Inset map highlighting testate amoebae sites from northeastern North America. (e) Inset map highlighting testate amoebae and pollen sites from Ireland.**

Several lines of evidence suggest that the 852/3 CE Churchill eruption occurred in winter, including the stratigraphic context of the tephra in proximal locations (West and Donaldson, 2000), the ash cloud trajectory (Muhs and Budahn, 2006) and the timing of ash deposition in Greenland. Cryptotephra from the eruption was identified in the NGRIP and NEEM-2011-S1 ice cores from northern Greenland in ice then dating to 847 CE based on the Greenland Ice Core 2005 (GICC05 chronology; Coulter et al., 2012; Jensen et al., 2014). Based on the revised NS1-2011 chronology (Sigl et al. 2015), the event is now dated to the winter of 852/3 CE (Fig 2), and is likely to have occurred between September 852 CE and January 853 CE, with sulfate deposition peaking in early 853 CE (Fig. 2e-f). The eruption also produced large quantities of ash and chlorine, the peak deposits of which are detected a few months prior to the sulfate peak in Greenland (Fig. 2). The NS1-2011 chronology is precise to the calendar year in 939 CE and 775 CE (Sigl et al., 2015) and it is therefore well-constrained over the time period of interest for this Churchill




eruption. The resultant conservative age uncertainty associated with the 852/853 CE Churchill eruption is winter
852/853 CE ± 1 calendar year.
Large volcanic eruptions have been implicated in global to hemispheric climate change and societal impacts (e.g.
Sigl et al., 2015; Stoffel et al., 2015; Büntgen et al., 2016, 2020; Oppenheimer et al., 2018; McConnell et al., 2020)
and raise the question of whether the Churchill eruption – amongst the largest magnitude eruptions of the Common
Era – also had a far-reaching impact. While extratropical eruptions are often thought to have less impact on climate
than tropical eruptions, recent modelling experiments have shown that large extratropical eruptions with injection
heights above ~17 km can have a significant hemispheric climate impact (Toohey et al., 2019). The Churchill
eruption certainly reached stratospheric heights, but it appears associated with only limited sulfate deposition in
Greenland ice cores (Fig. 2e), on the basis of which it is estimated to have produced 2.5 Tg of sulfur (ca. 5 Tg $SO_2$
(Toohey and Sigl, 2017)). This sulfate production estimate of the 852/3 CE Churchill eruption is an order of
magnitude less than the Alaskan 43 BC eruption of Okmok (McConnell et al., 2020), which was one of the three
largest eruptions, in terms of estimated aerosol forcing, of the last 2500 years (Sigl et al., 2015) and is less than a
third of the amount of sulfate produced during the 1991 eruption of Mount Pinatubo (Guo et al., 2004). The 852/3
Churchill eruption therefore provides a test case for investigating whether the event had the potential to impact
climate and society on the basis of the moderate estimated volcanic emissions, and the degree to which paleoclimate
reconstructions and historical records demonstrate environmental changes that might be regarded as consequences of
the eruption.
Given the extent of the Churchill WRAe isochron in glacial and terrestrial environments spanning North America
and western Eurasia, our study serves dual purposes. Our first aim is to examine potential NH impacts of the 852/3
CE Churchill eruption on climate, terrestrial environments and societies, using modelled forcing data, climate
simulations, palaeoenvironmental reconstructions and historical records. Our second aim is to use the WRAe tephra
isochron as a pinning-point between inter-continental paleoenvironmental records to characterize and compare
regional expressions of climate change near the outset of the Medieval Climate Anomaly (MCA), a period of
increased temperatures ca. 950–1250 CE (Mann et al., 2008; 2009). The WRAe isochron from the 852/3 CE
Churchill eruption is therefore aptly placed to identify leads and lags in MCA climate responses and improve
characterizations of the spatial and temporal extent of this warm period. We similarly use the tephra isochron to
critique the timing of land-use practices, inferred from pollen records, during a period of known societal
reorganisation, to determine the extent to which climate change played a role in socio-economic transformation.



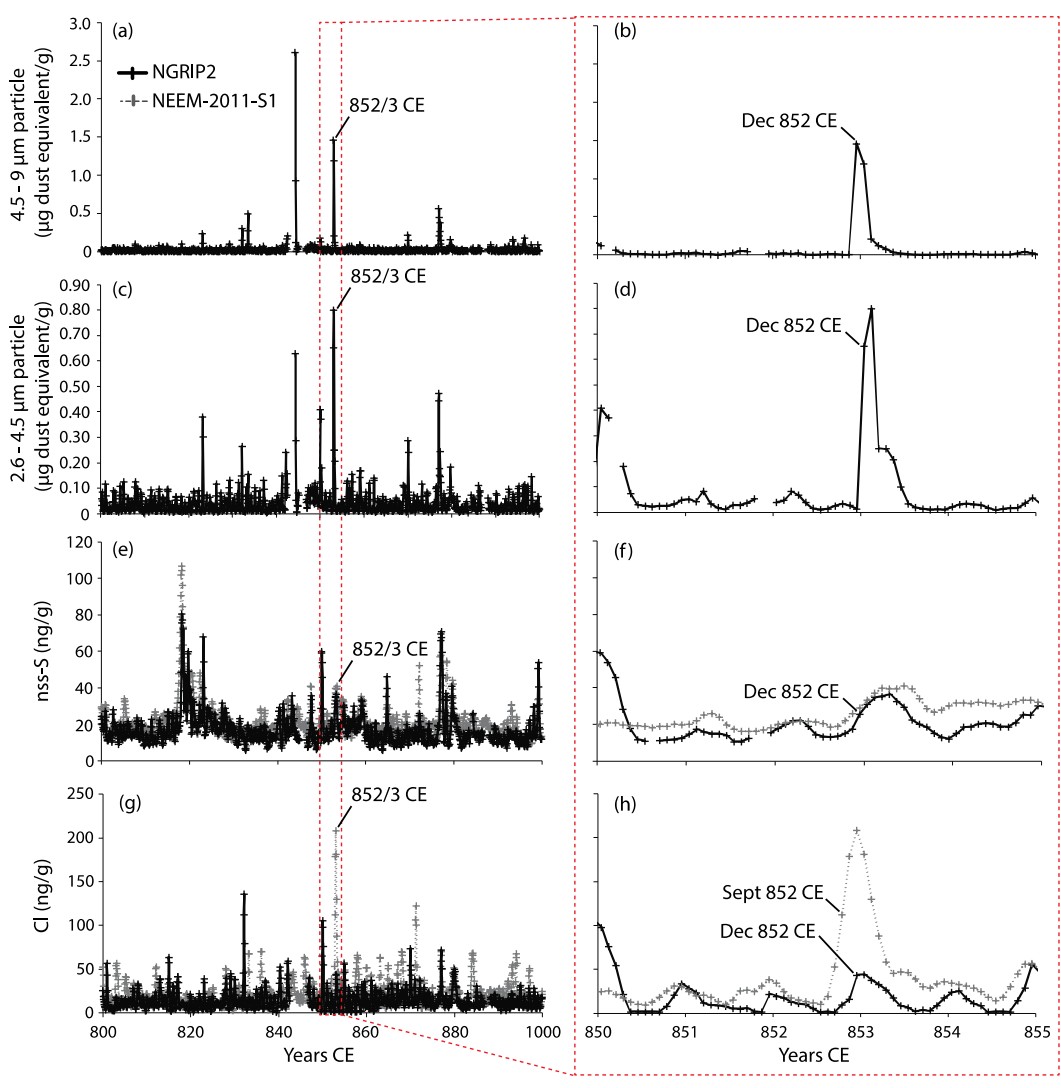


**Fig. 2: Geochemical characteristics of the 852/3 CE Churchill eruption based on concentrations of (a-b) ash inferred from 4.5–9 μm particles, (c-d) ash inferred from 2.6–4.5 μm particles, (e-f) non-sea salt sulfate (nss-S), and (g-h) chlorine (Cl), from Greenland ice cores NGRIP2 and NEEM-2011 S1 (Jensen et al., 2014) on the NS1-2011 chronology (Sigl et al., 2015).**



## 2. Methods

### 2.1 Revised eruption volume estimate and magnitude

Despite the considerable magnitude of the eruption that deposited WRAe, there has not been a spatially consistent estimate of its volume or magnitude using established methods (e.g., two-piece exponential function, Pyle 1989; Weibull function, Bonadonna and Costa, 2012). The most recent volume estimate for WRAe (Lerbekmo, 2008) used disparate isopach maps for the proximal and distal regions of the deposit and the uncertainty assessment was limited. Here we construct an updated isopach map for WRAe using a GIS-based synthesis of Lerbekmo's distal and proximal isopachs $\geq 0.5$ cm (Lerbekmo, 1975, 2008; Fig. 1a-b). We then calculate an updated tephra volume estimate by assuming deposit thinning follows a two-piece exponential function (Pyle, 1989; Fierstein and Nathenson, 1992). Dense rock equivalent (DRE) is calculated assuming a representative deposit density of 1.19 kg $m^{-3}$ and a dense rock density of 2.5 kg $m^{-3}$ (following Lerbekmo, 2008). These estimates of WRAe volume are the first to assess function-fitting confidence, allowing the mathematical model to account for the uncertainty of the deposit volume, especially < 0.5 cm.

### 2.2 Reconstructed forcing potential: stratospheric aerosol optical depth and radiative forcing

We develop a primary forcing reconstruction for the 852/3 CE Churchill eruption using the EVA(eVolv2k) 550 nm stratospheric aerosol optical depth (SAOD) reconstruction (Toohey and Sigl, 2017). Detailed explanations of the model selection and set-up are provided in Appendix A. We also generate a second SAOD reconstruction using the EVA_H model, which is an extension of the Easy Volcanic Aerosol Model (EVA, Toohey et al., 2016), that accounts for the $SO_2$ injection latitude and altitude and is calibrated using a more extensive observational dataset than EVA (Aubry et al., 2020). The EVA_H reconstruction uses the same $SO_2$ mass as EVA, the latitude of Churchill (61.38°N), and an injection altitude of 31.5 km. The injection altitude is based on the isopleth-derived top height estimate of 40–45km from Lerbekmo (2008) corrected by a factor of 0.725 to be representative of the altitude of the spreading umbrella cloud instead of the cloud top (Carey and Sparks, 1986). We also provide a 95% confidence interval on EVA_H prediction that accounts for uncertainties in model parameter (Aubry et al., 2020), the $SO_2$ mass uncertainty ($5 \pm 2.5$ Tg $SO_2$, Toohey and Sigl, 2017), and an assumed uncertainty of 30% on the injection height.

### 2.3 Climate model simulation

Climate conditions were simulated using the Community Earth System Model version 1.2.2 (CESM; Hurrell et al., 2013). The ensemble simulation consists of 20 ensemble members performed to study the impacts of the 852/853 CE Churchill eruption on climate. To generate the ensemble members, initially a seamless transient simulation is run from 1501 BCE (Kim et al., 2021) with time-varying orbital parameters (Berger, 1978), total solar irradiance (Vieira et al., 2011; Usoskin et al., 2014, 2016), greenhouse gases (Joos and Spahni, 2008; Bereiter et al., 2015), and volcanic forcing from the HolVol v.1.0 (Sigl et al., 2021) and eVolv2k (Toohey and Sigl, 2017) databases. The



necessary prescribed spatio-temporal distribution of volcanic sulfate aerosol for the simulation is generated using the
EVA Model (Toohey et al., 2016) and follows the same procedure employed by McConnell et al. (2020) and Kim et
al. (2021). The simulations used for the analysis have the spatial resolutions of approximately $1.9° \times 2.5°$ for the
atmosphere and land, and $1° \times 1°$ for the ocean and sea ice. The vertical grids use 30 levels for the atmosphere, 60
levels for the ocean and 15 levels for the land. The output data are provided at a monthly resolution. More details of
the simulations investigating the impact of the 852/3 CE Churchill eruption on climate are provided in Appendix B.
The anomalies of temperature and precipitation are calculated by subtracting the 845–859 CE multi-year monthly
means from the values at each grid point for the initial condition ensemble simulation. From these anomalies, the
seasonal means of each individual ensemble simulation are computed as well the ensemble means of 20 member
simulations. NH summer conditions reported here refer to climate conditions of June-July-August (JJA), and winter
conditions refer to December (of the previous year)-January February (of the reported calendar year) (DJF).
To test the statistical significance of changes in temperature and precipitation after the 852/3 CE Churchill eruption,
we use the Mann-Whitney U-test (for an example, refer to Kim et al., 2021) that compares the distributions of two
variables between the pre-eruption period (845–852 CE) and each individual after-eruption year (853, 854, and 855
CE). More details of the procedure for the significance tests are provided in Appendix B. In addition, the variability
of the spatially-averaged ensemble means of temperature and precipitation is compared with the pre-eruption
ensemble by assessing whether the variability falls within the range of two standard deviations from the means of
the pre-eruption period.
**2.4 Northern hemisphere (NH) tree-ring summer temperature and drought reconstructions**
NH summer (JJA) temperatures in the 850s CE were reconstructed using 13 NH tree-ring width and 12 maximum
latewood density chronologies (Guillet et al., 2017, 2020). Full details of the nested principal component regression
(PCR) used to reconstruct NH JJA temperature anomalies (with respect to 1961–1990) and the model calibration are
provided in Appendix C. To place the summer temperature anomalies within the context of climate variability at the
time of major volcanic eruptions, we filtered the final reconstruction by calculating the difference between the raw
time series and the 31-year running mean. Further investigation of volcanic-forced cooling was facilitated by
filtering the original reconstructions using a 3-year running mean to filter out high-frequency noise. To estimate the
spatial variability of summer cooling induced by the winter 852/3 CE eruption, we also developed a 500–2000 CE
gridded reconstruction of extratropical NH summer temperatures (more details are provided in Appendix C).
Estimated soil moisture anomalies for the $9^{th}$ century are extracted from tree-ring reconstructions of the gridded
summer (JJA) Palmer Drought Severity Index (PDSI) over North America, Europe, and the Mediterranean (Cook et
al. 2010, Cook et al. 2015). The PDSI metric integrates the influence of both precipitation, evapotranspiration, and
storage on soil moisture balance throughout the year and is normalized so that values can be compared across regions
with a range of hydroclimate conditions. Positive values indicate anomalously wet conditions, while negative values



are anomalously dry for that location, and normal conditions are set to zero. Tree-ring PDSI reconstructions in North
American and Euro-Mediterranean Drought Atlases are developed using the point-by-point regression approach
described by Cook et al. (1999).
**2.5 Testate amoebae peatland water table depth (i.e. summer effective precipitation) reconstructions**
Testate amoebae are a well-established palaeoenvironmental proxy used to reconstruct past hydroclimatic variability
in ombrotrophic (rain-fed) peatlands because species assemblages predominantly respond to changes in peatland
surface moisture during summer months, and tests are preserved in the anoxic peat strata (e.g. Woodland et al.,
1998; Mitchell et al., 2008). For this study, testate amoeba analysis was completed on cores obtained from 11
ombrotrophic peatlands located in Maine (n = 2), Nova Scotia (n= 3), Newfoundland (n = 4) and Ireland (n = 2)
(Fig. d-e), in which the presence of the WRAe has been confirmed by electron probe microanalysis of the volcanic
glass (Swindles et al., 2010; Pyne O'Donnell., 2012; Mackay et al., 2016; Monteath et al., 2019; Jensen et al., in
press; Plunkett., unpublished; Appendix D). The peatland sampling approaches used here are outlined in Mackay et
al. (2021), and testate amoeba analysis was completed using standard protocols (Hendon and Charman, 1997; Booth
et al., 2010) across all cores at multidecadal resolution (approximately 40-years), equating to 2 to 4 cm intervals.
Testate amoebae were extracted from 1 cm$^3$ subsamples following standard procedures (Hendon and Charman,
1997; Booth et al., 2010). At least 100 individual tests were identified (Payne and Mitchell, 2009) per sample using
the taxonomy of Charman et al. (2000) and Booth (2008). Testate amoebae water table depth (WTD) reconstructions
were obtained using the tolerance-downweighted weighted averaging model with inverse deshrinking (WA-Tol inv)
from the North American transfer function of Amesbury et al. (2018). Reconstructed WTD values were normalised
for comparative purposes (Swindles et al., 2015; Amesbury et al., 2016). Two WTD reconstructions exist from
different coring locations on Sidney Bog, Maine (Clifford and Booth, 2013; Mackay et al., 2021); therefore, a
composite record was constructed based on interpolated average WTD values (Appendix E). Two WTD
reconstructions also exist from different coring locations on Saco Heath (Clifford and Booth, 2013; Mackay et al.,
2021), however, a composite record was not created for this site since one record contains a pronounced hiatus
below the WRAe horizon, relating to a burning event (Clifford and Booth, 2013). The Saco record presented within
this study contains no evidence of a hiatus until later in the record, ca. 1000 CE, when the accumulation rate
decreases (Appendix D). Core chronologies were developed using Bayesian analysis within the R package
"BACON" (Blaauw and Christen, 2011) based on $^{14}$C dates and tephrochronologies (Appendix D). Radiocarbon
dates were calibrated using the NH IntCal20 calibration curve (Reimer et al., 2020) and are reported as Common Era
dates.
**2.6 Pollen vegetation reconstructions**
The 9th century in Ireland was a time of significant socio-economic reorganisation and possibly population decline
(Kerr et al., 2009; McLaughlin et al., 2018; McLaughlin, 2020). To investigate the extent to which these events may
have been driven the effects of either the 852/3 CE eruption or the transition to the MCA, we compiled land-use



proxy data from five pollen records (Fig. 2e) that included the Churchill ("AD860B") tephra as a chronological tie-
point (Hall, 2005; Coyle McClung, 2012; Plunkett, unpublished data). Raw data were recategorized by biotope, with
a specific focus on the ratio of arboreal pollen (AP) to non-arboreal pollen (NAP), and the representation
(percentage of total dryland pollen) of taxa associated with pastoral or arable farming. Age-models were constructed
for each site based on tephrochronological and $^{14}$C dates in the same manner used for the testate amoebae records
(Sect. 2.5).
**2.7 Historical records**
A wide range of written sources were examined to collate the extant historical record of climate and weather for the
period 850–856 CE. This survey focused on Europe – northwestern insular Europe (Irish and Anglo-Saxon annals)
and continental Europe (annals and histories covering Byzantine, Carolingian and Umayyad lands) – southwest
Asia, North Africa (Abbasid and Byzantine texts), and Tang-era eastern China. To place the 852/3 CE eruption in a
wider context where effects of the eruption are apparent, we employ evidence for large subsistence crises
('famines') and seemingly more circumscribed crises ('lesser food shortages') spanning 800–900 CE reported in
Carolingian sources, which comprise one of the densest records of subsistence crises extant for the 9[th] century
anywhere (Newfield 2013, Devroey 2019).
**3. Results**
**3.1 Volume estimate and magnitude**
WRAe deposit bulk tephra volume was modelled as a mean value of 49.3 km$^3$, with an estimated 95% confidence
interval (CI) of 39.4–61.9 km$^3$. The deposit constituted a mean dense rock equivalent (DRE) volume of 23.6 km$^3$
(95% CI, 18.8–29.6 km$^3$ at 95% confidence) and weighed about 48.7 Gt (95% CI, 38.9–61.2 Gt at 95% confidence).
Such volumes and masses indicate the eruption that deposited WRAe was of volcanic explosivity index (VEI) 6 and
a magnitude (M) of around 6.7 (95% CI, 6.6–6.8 at 95% confidence).
**3.2 Climatic forcing potential of 852/3 CE Churchill eruption**
The EVA(eVolv2k) reconstructed stratospheric aerosol optical depth (SAOD) for the 852/3 CE eruption is relatively
moderate, with a peak aerosol optical depth perturbation of 0.049 in terms of global mean, and 0.078 in terms of
North Hemisphere (NH) mean (Fig. 3a-b). In comparison, the global mean SAOD following the Pinatubo 1991
eruption was 2–3 times larger (Thomason et al., 2018) and the reconstructed global mean SAOD for the largest
eruptions of the Common Era (Fig. 3a) reaches 0.3–0.6 (e.g., 0.56 for the Samalas 1257 CE eruption). For the 9[th]
century alone, four volcanic events have a peak global mean SAOD exceeding that of the 852/3CE Churchill
eruption. The EVA_H reconstruction (Fig. 3b), which accounts for the SO$_2$ injection latitude and altitude, suggests
an even smaller global mean SAOD perturbation of 0.033 (95% confidence interval 0.018–0.048). In terms of the
latitudinal distribution of the SAOD perturbation, both the EVA (Fig. 3c) and EVA_H (not shown) reconstructions





produce a SAOD perturbation that is much stronger in the NH but propagates to the tropics and Southern
Hemisphere. Based on the EVA(eVolv2k) SAOD estimate and using volcanic forcing efficiency from Marshall et al.
(2019), the global mean radiative forcing peaked at -0.92 W m$^{-2}$ (Fig. 2b), a value roughly one-third that for the
Mount Pinatubo 1991 eruption (e.g., Schmidt et al., 2018).

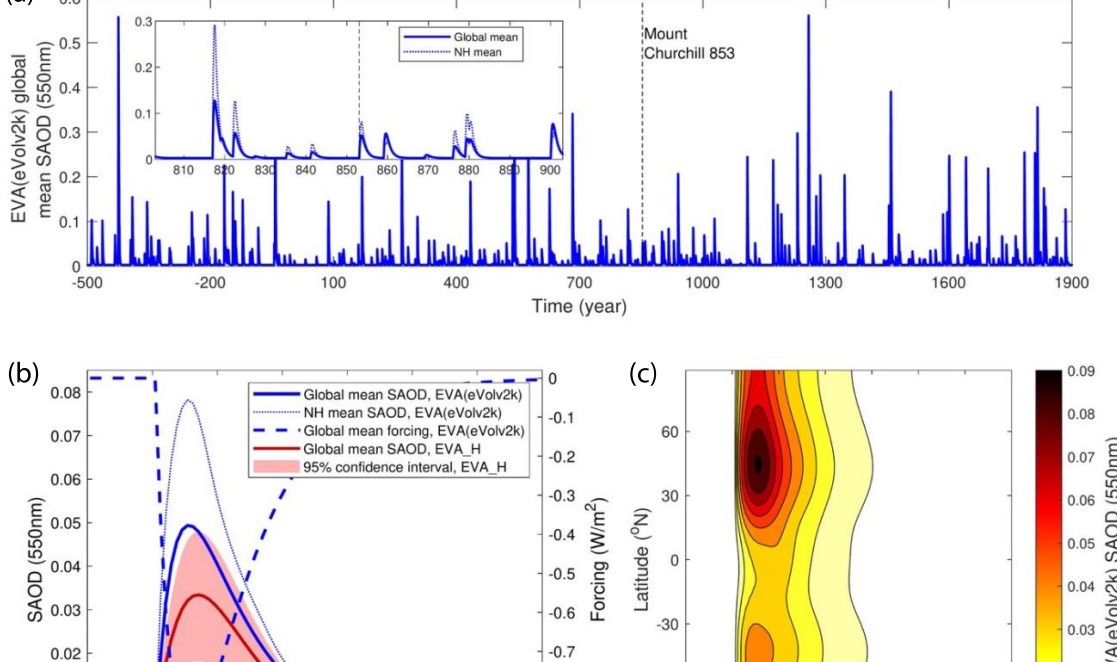


**Fig. 3: Stratospheric aerosol optical depth (SAOD, 550 nm) reconstructed for the Churchill 852/3 eruption. (a) The 500**
**BCE–1900 CE EVA(eVolv2k) reconstructed global mean SAOD, with the inset showing details for the 803–903 period**
**and both the global mean and North Hemisphere (NH) mean SAOD. (b) The same time series for the 852–859 CE period,**
**during which the Churchill 852/3 eruption is clearly seen. This panel also shows the global mean radiative forcing**
**reconstructed from the EVA(eVolv2k) SAOD, and an alternative SAOD reconstruction using the EVA_H model, an**
**extension of EVA that accounts for the SO$_2$ injection and latitude for reconstructing global mean SAOD. (c) Time-latitude**
**evolution of SAOD as reconstructed with EVA(eVolv2k).**
**3.3 Annually resolved climate reconstructions**
**3.3.1 NH tree-ring-based climate reconstructions**
NH summer temperature reconstructions based on tree-ring records reveal long-term decadal-scale temperature
fluctuations between 500–2000 CE (Fig. 4a). All tree-ring based NH JJA reconstructions contain a short-lived
decreasing temperature trend from 851 CE that peaks in 853 CE, with temperatures anomalies (relative to 1961–

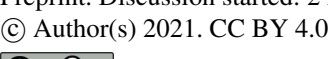



1990) reaching –0.85°C in the filtered reconstruction (Fig. 4b). The 1961–1990 reference period used for the tree-
ring reconstructions was 0.1°C warmer than the modelled climate simulation reference period (845–852 CE). Cold
temperatures persist in 854 CE (with -0.65 and -0.5°C in the filtered and unfiltered reconstructions, respectively),
before attaining pre-eruption levels in 855 CE (Fig. 4b). The cold temperature anomaly observed in 853 CE is
significant and among the 5th percentile of coldest values in the filtered and unfiltered reconstructions and very close
to the 1st percentile of the coldest values in the distribution of the filtered reconstruction (Appendix F). Over the
period 500–2000 CE, 853 CE ranks as the 28th and 18th coldest events in the unfiltered and filtered reconstructions,
respectively (Fig. 4a). Further investigation of volcanically forced cooling as examined using a 3-year running mean
places 853-856 CE as the 11th coldest 3-year period between 500 and 2000 CE (Appendix G). An examination of the
30 coldest 3-year periods from the filtered time series highlights that all such periods are preceded by an eruption or
a group of eruptions, and 19 of these eruptions occur within two years before the ranked cold periods  (Appendix G).
Spatial patterns of the hemisphere-wide JJA cooling in the early 850s are complex (Fig. 5a): generally cold
conditions prevailed over western and central Europe as well as Scandinavia (anomalies exceeding –0.8°C with
respect to the 1961–1990 mean) – and to a lesser extent Alaska (with peak cooling in 854 CE) – between 851 and
854 CE. The peak cooling in the NH in 853 and 854 CE seen in Fig. 4b is explained by the strong cooling of Central
Asia and vast parts of Siberia in the same years (Fig. 5a). While clear warming is evident in central and western
Europe and Scandinavia in 855 CE, low temperatures persist in Central Asia in 855 CE.
Summer PDSI reconstructions based on tree-ring records reveal a shift from wet to drier conditions in parts of
western Europe in 854 CE which persists into 855 CE (Fig. 5d). Wetter conditions in 853 CE in northern Europe and
dry anomalies in North Africa and parts of the Mediterranean are potentially indicative of a positive phase of the
North Atlantic Oscillation. By 855 CE, dry conditions in northern and western Europe and, in 855 CE in the eastern
United States are more similar to the pattern expected during a negative phase of the North Atlantic Oscillation
(Anchukaitis et al. 2019).  Eastern North American tree-ring moisture reconstructions however are also consistently
dry from 852 through 855 CE. Tree-ring records in the western half of the continent reveal a mixed PDSI anomaly,
generally indicating wetter conditions to the southwest and drier in the northwest, reminiscent of the moisture
anomalies during a El Niño event in the tropical Pacific (Fig. 5d).

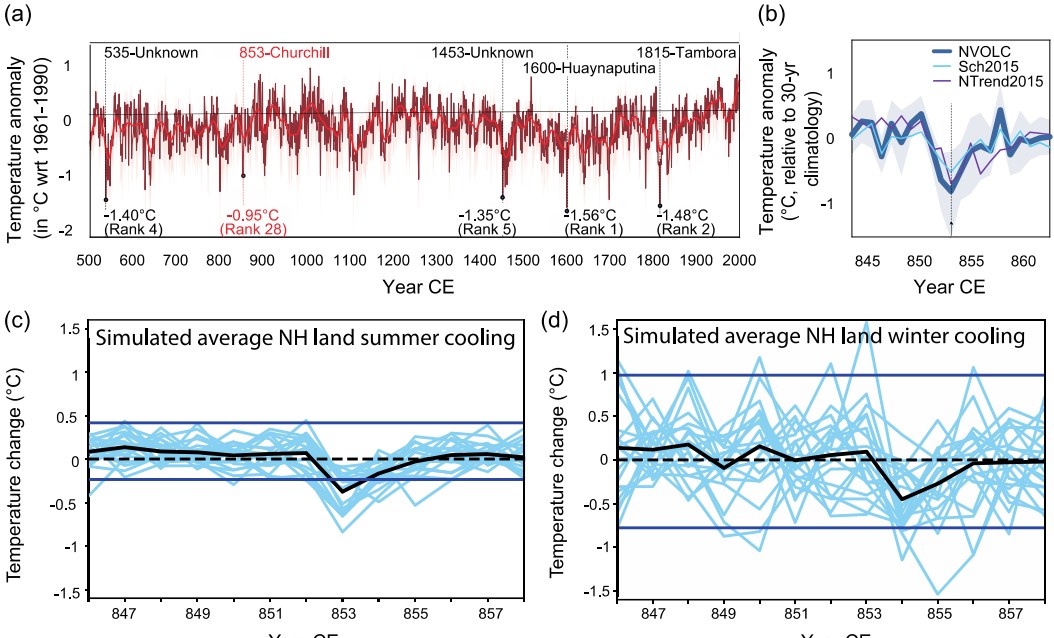

**Fig. 4 (a-b) Tree-ring-derived temperature reconstructions around the time of the 852/3 CE Churchill eruption: (a) Unfiltered NH extra-tropical land (40–90°N) summer temperature anomalies (with respect to the period 1961–1990) since 500 CE. The red lines represent the interannual temperature anomaly variations and the grey lines represent the 95% bootstrap limits. (b) Comparison of the NVOLC cooling observed after the winter 852/3 CE eruption in the reconstruction filtered with a 31-yr running mean with other NH reconstructions, Sch2015 (Schneider et al., 2015) and N-TREND2015 (Wilson et al., 2016). Grey shaded area represents the uncertainty associated with NVOLC temperature anomaly reconstruction. (c-d) Simulated NH climate before and after the 852/3 Churchill eruption: (c) 846–858 CE time series of spatially-averaged NH extratropical (15–90°N latitudes) land temperature anomalies from 20 ensemble simulations for summer (JJA), and (d) winter (DJF) in light blue lines. The thick black lines indicate the ensemble means, and the horizontal blue lines represent two standard deviations from the ensemble means of the 845–852 CE pre-eruption period. For comparative purposes, the 1961–1990 reference period used for the tree-ring reconstructions (a-b) was 0.1°C warmer than the modelled climate simulation reference period (c-d) of 845–852 CE.**

**3.3.2 NH modelled climate scenarios**

Simulated summer (JJA) temperature anomalies derived from the CESM reveal a widespread cooling in the NH extratropical regions in 853 CE that reaches an ensemble mean value of approximately –0.29 °C (Fig. 4c). In many extratropical regions, the decrease in summer temperature is statistically different to the pre-eruption period at a 5% confidence level (Fig. 5b) and the ensemble means of the temperature anomalies in summer 853 CE are greater than two standard deviations from the 845–852 CE pre-eruption period mean, placing it among the 2[nd] percentile of the coldest simulated temperatures. Cool conditions mainly persist into 854 CE, albeit with smaller temperature anomalies (NH land average cooling of –0.15°C; Fig. 4c), but by 855 CE, warm temperature anomalies return, for example, to parts of southeast Europe, northeast Canada and the North Pacific. Modelled winter (DJF) temperature anomalies reveal a cooling trend that starts and peaks approximately –0.32°C in 854 CE and recovers by 856 CE (Fig. 4d). Modelled winter (DJF) temperature anomalies reveal a hemispheric mean, ensemble mean cooling anomaly that peaks



at approximately –0.32°C in 854 CE and recovers by 856 CE (Fig. 4d). The ensemble mean winter cooling in 854 CE
is more spatially variable than the 853 CE summer cooling, with warm temperatures anomalies persisting in parts of
Scandinavia, central Europe and western North America during the winter months (Fig. 5c). The variability among
the ensemble members during the after-eruption period (853–855 CE) is high, with the NH land surface temperature
means ranging from –0.84 to 0.25°C in summer and –1.54 to 1.57°C in winter.
The modelled summer (JJA) and winter (DJF) precipitation anomalies vary spatially and temporally between 851–
855 CE (Fig. 5e-f), although the post-eruption variability of precipitation is statistically indistinguishable from that
of the pre-eruption period. Parts of western Europe show slightly drier conditions in winter 853 CE, with wetter
conditions prevalent in western Scandinavia. The summer of 853 CE is characterised by slightly wetter conditions in
parts of western Europe (Fig. 5e). The spatially-averaged ensemble mean of precipitation indicates that all variation
occurs within one standard deviation of the pre-eruption period means (Appendix H); there is therefore no obvious
statistical differences between modelled summer and winter precipitation patterns associated with the 852/853 CE
Churchill eruption in the NH.







**Figure 5:  Reconstructed and simulated NH spatial patterns of temperature and precipitation anomalies (a) Growing season gridded (1° lat × long) temperature anomalies reconstructed over the NH (40–90° N) between 851 and 855 CE based on tree-ring reconstructions. Scale extends from red, representing a temperature increase, to purple, representing a temperature decrease. (b) Annually-averaged ensemble means of simulated temperature anomalies for summer, and (c) winter. (d) Spatial patterns of boreal summer (June–August) Palmer Drought Severity Index (PDSI) anomalies (Cook et al. 2010, Cook et al. 2015). The PDSI scale extends from blue, representing wetter-than-normal conditions at that location, to brown, representing drier-than-normal conditions. (e) Annually-averaged ensemble means of simulated precipitation anomalies for summer, and (f) winter. Dotted regions in (b), (c), (e) and (f) indicate where the changes are statistically significant (based on the Mann-Whitney-U-test) compared to the pre-eruption period.**

**3.4 Multidecadal scale palaeoenvironmental reconstructions**

**3.4.1 Peatland hydrological change associated with WRAe deposition**

The compilation of WTD data in peatlands indicates no consistent response at the time of the WRAe deposition (Fig. 6). Both Irish peatlands record wet conditions relative to the preceding decades at the time of WRAe deposition, but the Dead Island record indicates a subsequent long-term drying whilst Cloonoolish records a temporary drying before a shift to wetter conditions. Two of the three peatlands in eastern Newfoundland record wetter conditions following the WRAe deposition. Jeffrey's Bog in southwestern Newfoundland and the peatlands in Nova Scotia become drier following the eruption but the duration and magnitude of the water table lowering vary between peatlands. For example, the longer-term drying trends in Jeffrey's Bog, Framboise Bog and Villagedale Bog persist over approximately 200 years whilst the drying in Petite Bog is less pronounced and shorter-lived (ca. 50 years). The peatlands in Maine register a temporary shift to wet conditions following the WRAe deposition.

Although most of the sites reflect centennial-scale trends in WTD, the higher temporal resolution of Petite and Cloonoolish bogs (11 and 12.5 years respectively) allow decadal-scale responses of the peatlands following the eruption to be considered. Each bog experienced a short-term change towards drier conditions before returning to the prior trend to wetter conditions, but the scale of each hydrological shift lies within the levels of variability of the WTD records.

**3.4.2 Peatland hydrological change during the Medieval Climate Anomaly**

We find no consistent MCA signal registered in the peatland WTD reconstructions (Fig. 6). Our peatland WTD records indicate that the medieval period was characterised by variable hydrological conditions. The onset of changes towards drier conditions, which may signal the warm Medieval Climate Anomaly, varies temporally and spatially. The earliest dry shift starts ca. 900 CE in northern Nova Scotia (Framboise Bog) and some records from Newfoundland (Jeffrey's Bog and Nordan's Pond Bog), whilst this hydroclimatic change is registered ca. 100 years later in records from southern and central Nova Scotia (Villagedale Bog, Petite Bog), Maine (Sidney Bog) and Ireland (Cloonoolish). All records in this study register temporary wet shifts at approximately 850 CE and 1050-1150 CE, although the extent and durations of the wet shifts vary. The presence of the WRAe isochron conclusively demonstrates that the onset of the wet shift ca. 850 CE is not synchronous. There is also a high degree of spatial variability between records from sites proximal to one another, with some recording contradictory hydrological



393    conditions, such as Saco Heath and Sidney Bog in Maine and Nordan's Pond Bog, Pound Cove Bog and Southwest

394    Pond Bog in Newfoundland.

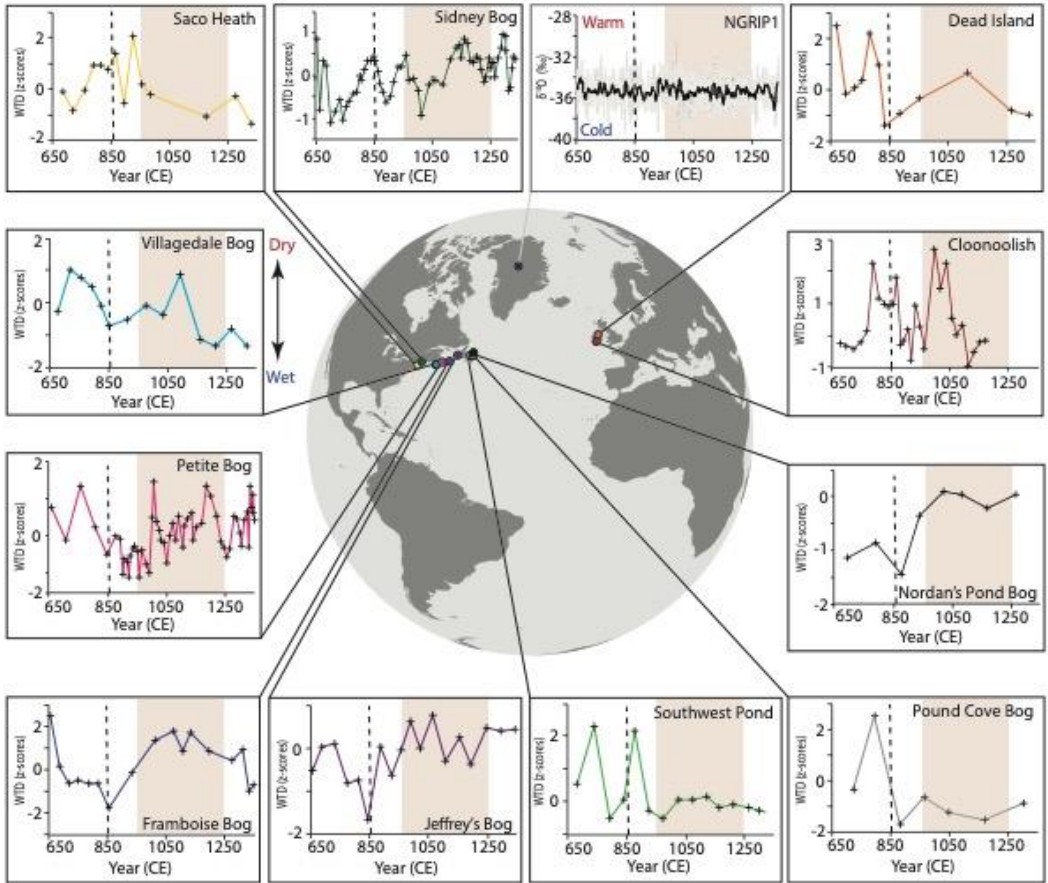

395

**Fig. 6: Available moisture reconstructions from terrestrial and glacial archives containing the WRAe from the North Atlantic region. Records have been developed using δ18O isotopes from NGRIP1 (Vinther et al., 2006), where the black line represents decadal-scale moving average and data are plotted on NS1-2011 chronology (Sigl et al., 2015; for detailed 9th century NGRIP1 δ18O isotopes see Appendix J), and peatland water table depths inferred from testate amoebae. Dashed vertical line represents the WRAe and the pink shaded box represents the MCA time period based on the Mann et al. (2009) timings (950–1250 CE). Sites have been arranged clockwise with Irish records (Dead Island and Cloonoolish) located on the top right of the diagram followed by North American sites from north-east to south-west with records from Newfoundland (Nordan's Pond Bog, Pound Cove Bog, Southwest Pond and Jeffrey's Pond), Nova Scotia (Framboise Bog, Petite Bog, Villagedale Bog) and Maine (Saco Heath and Sidney Bog) and finally the Greenland NGRIP1 record (top of the diagram).**



### 3.4.3 Vegetation reconstructions

Pollen records from Ireland show considerable variability in the intensity and extent of farming (Fig. 7). The WRAe deposition from the 852/3 CE eruption coincides with the pinnacle of land clearance (reduced arboreal pollen) in central Ireland (Clonfert and Cloonoolish bogs), after which pastoral and arable indicators start to decline as woodland expands. Sites in the northeast of Ireland show less coherent trends than those in central Ireland. At Garry Bog, arable weeds temporarily dip at the time of the eruption, although cereals are still evident. In contrast, evidence for farming is very limited at nearby Frosses Bog, highlighting the localised nature of land use in the vicinity of Garry. At Lake View, moderate levels of farming are recorded, and these increase slightly following the eruption before a decline in activity begins later in the century. The spatial diversity in the pollen records (even within a single region) demonstrates that changes in land-use in the 9th century cannot be attributed to any one environmental trigger, and very likely reflect differences in local-to-regional economic organisation and demographic pressures.

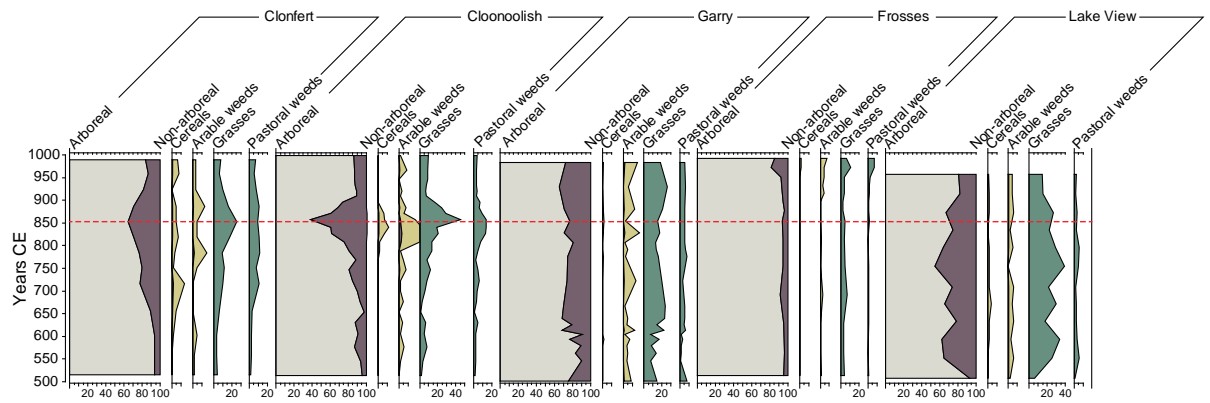

**Fig. 7: Summary pollen records from five sites in Ireland, showing the ratio of arboreal to non-arboreal (dryland) taxa and indicators of arable and pastoral environments. Cereal and arable weed curves are shown with a ×10 exaggeration. The red dashed line indicates WRAe.**

### 3.5 Historical records

Historical records from Europe characterize the 850s CE as time of climate instability (Table 1). Carolingian sources observe severe winter flooding in western Germany in 849-850 CE, and severe summer heat, drought and a *fames* (food shortage) in 852 CE (Newfield, 2010; Haldon et al., 2018). Immediately following the 852/3 CE eruption, there is contemporary evidence for a severe famine – such that horse flesh was eaten – which the Annals of Xanten specify took place in Saxony in 853 CE, though beginning possibly in 852 CE (Newfield, 2010; Haldon et al., 2018). The eyewitness annalist of the Annals of Fulda noted that in 855 CE, 'unusually changeable weather brought loss to many through whirlwinds, storms and hailstorms'. The Annals of St Bertin describe the winter of 856 CE as severe and dry, being also accompanied by a severe epidemic, 'which consumed a great part of humanity' (Newfield, 2010). Heavy snowfall is reported in Ireland for 23 April 855 CE, with extreme cold implied by frost and





load-bearing ice across the winter of 855/6 CE and 856 CE was also deemed a tempestuous and harsh year in
Ireland. A severe windstorm occurred in 857 CE and autumn weather in 858 CE characterised as wet and destructive
to agriculture in Ireland. A potentially less reliable source (the Fragmentary Annals of Ireland) also reported a
famine in the autumn of 858 CE (Ludlow, 2010; Ludlow et al., 2013). The Xanten annalist recorded a great
epidemic in 857 CE in northwest Germany, causing 'swelling bladders' (or 'swelling tumours') and 'festering sores'
that putrefied limbs (Newfield, 2010). While disputable, this epidemic has long been identified as one of ergotism
(Hirsch, 1885; Duby, 1974), which is caused by ingestion of the ergot fungus of rye and other grains and is more
common in cold and wet growing seasons (Kodisch et al., 2020). The St Bertin annalist reported the epidemic in 858
CE. That year too, in May, such a heavy rain fell that the river Meuse burst its banks, flooding Liege (present-day
Belgium) and tearing down buildings (Table 1).
A wider chronological consideration of the Carolingian evidence reveals that food shortages occurred in several
other decades of the 9th century in Carolingian Europe (Fig. 8). This observation reinforces the point that a
correspondence (or near-correspondence) between the dating of the Churchill eruption and the documented events of
the 850s CE certainly does not confirm a causal linkage. Some food crises of the 9th century were, moreover, vast
and longer lasting than those (reliably) documented here for the 850s, with the Carolingian sources also observing
widespread crises associated with climate anomalies in the 820s, 860s and 870s (Newfield, 2013; Haldon et al.,
2018; Devroey, 2019). One mid-10th-century source does observe a *hiemps gravissima* (gravest winter) preceding a
five-year *fames intolerabilis* (intolerable food shortage) vaguely datable to the early 850s and possibly located in
and beyond northern France and Belgium (Newfield, 2013). However, this evidence must be treated with caution
given its non-contemporaneity, unsecure dating of events and dramatized tone. It can be noted that the written record
of food shortages is certainly incomplete for this period of European history, such that some events of the 850s may
have gone undocumented. We may also be posit that if extreme weather did not occur when it could affect harvests
sufficiently to trigger a serious subsistence crisis, or when society otherwise proved resilient (e.g. through adequate
stored reserves), it may have been deemed less relevant for recording.

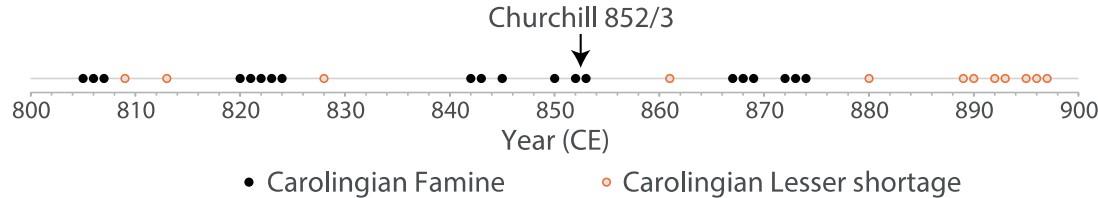


**Figure 8: 9th century reports of large subsistence crises ('famines', black) and seemingly more circumscribed crises ('lesser food shortages', orange) recorded in Carolingian sources (Newfield, 2013). Note again that the record of food shortages is imperfect: some crises may not have been recorded and the extent and severity of several recorded crises are difficult to determine.**





**Table 1: Climate-relevant events recorded in Irish, Carolingian, Anglo-Saxon, Byzantine, Italian, Iberian, Abbasid and**
**Egyptian sources between 850-858 CE. Locations given reflect where the texts were likely at the time compiled, though**
**the phenomena recorded could have been more widespread. Cases were the phenomena locations are instead given are**
**denoted by *.**

| Year CE | Event | Location | Source |
|---------|-------|----------|--------|
| **850** | Flooding | Rio Guadalquivir | *Meklach et al. (2021)* |
| **850** | Food shortage | western Germany | *Annals of Fulda* |
| **850** | Winter flood, excessive summer heat | western Germany | *Annales Xanten* |
| **851** | Low summer flood | Nile (mainly Blue Nile, rising in Ethiopian highlands) | *Kondrashov et al. (2005)* |
| **852** | Excessive heat contributing to a food shortage | northwestern Germany | *Annals of Xanten* |
| **853** | Food shortage | northwestern Germany | *Annals of Xanten* |
| **854/5** | Cold winds, disease | Baghdad | *Ibn al-Jawzi* |
| **855** | Deep snow in late April | * Ireland (unspecific) | *Annals of Ulster* |
| **855** | Frost and frozen lakes (to loadbearing strength) | * Munster, Ireland | *Annals of the Four Masters, Fragmentary Annals* |
| **855** | Large hail | Baghdad | *Ibn al-Jawzi* |
| **855** | Unusual hail and storms | central Germany | *Annals of Fulda* |
| **855/6** | Lakes and rivers frozen | * All Ireland (implied) | *Annals of Ulster, Chronicon Scotorum, Annals of the Four Masters, Fragmentary Annals* |
| **856** | Tempestuous and harsh year | * Ireland (unspecific) | *Chronicon Scotorum, Annals of Ulster* |
| **856** | Severe, dry winter, epidemic | northern France | *Annals of St Bertin* |
| **857** | Lightning kills three persons | * Meath, eastern Ireland | *Chronicon Scotorum, Annals of Ulster, Annals of the Four Masters, Fragmentary Annals* |
| **857** | Great windstorm, destroys trees and lake islands (crannogs) | * Ireland (unspecific) | *Annals of Ulster* |
| **857** | Epidemic | northwestern Germany | *Annals of Xanten* |
| **858** | Epidemic, flood | northern France | *Annals of St Bertin* |
| **858** | Wet autumn, destructive to agriculture and/or fruiting plants | * Ireland (unspecific) | *Annals of Ulster* |
| **858** | Famine | * Ireland (unspecific) | *Fragmentary annals* |
| **858** | Epidemic, heavy rain and snow | Baghdad | *Ibn al-Tabarī, Ibn al-Jawzi* |


Elsewhere, in Iberia, we read only of significant flooding along the Rio Guadalquivir in 849 and 850 CE (Meklach
et al., 2021), while there are no known reports in Anglo-Saxon, Byzantine, Italian or Iberian sources of anomalous
weather ~853 CE or potentially related societal events that might suggest climate perturbations then. Further east,
however, the scholar Ibn al-Jawzi (writing in the twelfth century, but with access to contemporary sources for our





period) wrote that for the year 240 (854/5 CE), a cold wind 'came out from the land of the Turks and many died
from having a cold… the winds continued to Iraq and the people of Samarra and Baghdad suffered from fever and
cough and cold.' Then, in March 855 CE, 'massive hail' fell in Baghdad, 'sized larger than nuts, along with heavy
rainfall.' In the year 244 (858/9 CE) the eyewitness scholar, al-Tabarī recorded that in Syria, 'pestilence broke out
(and the reason for that was) that the air was cold and full of dew, the rainfall heavy… prices rose and there was
snow'. When al-Jawzi later wrote up this report in his own history, he added that the snow lasted more than two
months (Table 1).
A suppression of the East African Monsoon may have been expected with a NH winter eruption of this magnitude
(Manning et al. 2017; Oman et al. 2006); however, the extant historical sources do not identify such a happening.
Egyptian historical records are silent. The Nilometer recordings of high and low stands of the annual Nile flood at
and after the date of the eruption do not appear anomalous, and there are no known incidences of food crisis or
famine that we would otherwise expect from unusually low Nile flooding (Hassan, 2007). Only a low flood (the 5th
lowest of the 9th century) is recorded in 851 CE (Kondrashov et al., 2005). Given that this would have largely been
the product of lessened monsoon rainfall in summer over the Ethiopian highlands, this low Nile cannot be credibly
linked to the eruption date of Mt Churchill, even when accounting for a +/- 1 year potential uncertainty that places
the potential eruption date as early as winter 851 CE. There are also no sources from the Nubian Nile that suggest
the presence of climatic anomalies or societal reaction to them during this time period (Adam Laitar and Giovanni
Ruffini, pers. comm.).
Chinese historical sources register local and regional weather anomalies and impacts in eastern China in the years
following the eruption. Of particular note is a drought in the summer of 852 CE, affecting the Huainan Circuit,
comprising some 12 prefectures and 53 counties, and situated between the Huai and the Yangzi rivers. Famine
associated with the drought induced migration, with people resorting to wild foods (Zhang, 2004; as per the *New
Book of Tang*). There is also a record in 854 and 855 CE of a further drought followed by a famine in several
counties in Huainan (Chen, 1986; Zhang, 2004). The government intervened with relief measures consisting of tax
reductions and food shipments. A devastating flood then occurred in 858 CE and engulfed a large area, including
several prefectures along the Grand Canal, in Hebei, Henan and Huainan circuits (Somers, 1979; Chen, 1986;
Zhang, 2004). Water then rose several feet, causing massive loss of life. Given its immense geographical area,
natural disasters were common on at least local scales at various latitudes across the Chinese landmass, which,
coupled with human disruptions, such as banditism and government neglect, often had calamitous social and
economic effects. We can therefore again stress that the events of the 850s CE cannot be uncritically linked to the
climatic impacts of the Mt Churchill eruption, and when considered in the context of 9th-century Chinese climate
history, more severe, widespread and prolonged weather extremes are documented (Yin et al., 2005). However, the
reporting of drought in 852, 854 and 855 CE in Huainan (east China) is consistent with expectations of a
suppression of East Asian summer monsoon rainfall following a high-latitude Northern Hemispheric eruption (e.g.,
Zhuo et al., 2014; Iles and Hegerl, 2015).



## 4. Discussion

### 4.1 Climatic impact of the 852/3 CE Churchill eruption

The VEI 6 eruption of Churchill in the winter of 852/3 ± 1 CE was amongst the largest eruptions of the Common Era and dispersed ash eastwards over a distance of 7,000 km. Despite its large magnitude, on the basis of sulfate deposition in Greenland ice cores, the eruption appears to have had only moderate climate forcing potential: the SAOD perturbation is concentrated in the NH and there are four other volcanic events in the 9[th] century that have larger global mean SAOD. The 852/3 CE Churchill eruption therefore contributes to the known examples of large magnitude Common Era eruptions that are associated with moderate atmospheric sulfate burdens as reconstructed from ice-cores (Sigl et al., 2013, 2014, 2015), such as Taupo 232 ± 10 CE (Hogg et al., 2012; Hogg et al., 2019); Changbaishan 946 CE (Sun et al., 2014; Oppenheimer et al., 2017); and Long Island 1661 ± 10 CE (Blong et al., 2018).

Despite the moderate climate forcing potential of the 852/3 CE Churchill eruption estimated from ice core sulfate records, there is evidence for a strong NH cooling associated with 853 CE. Tree-ring temperature reconstructions show temperature declines centred on summer 853 CE with a peak magnitude of around –0.8°C. In terms of 3-year mean NH summer temperature, the 853-856 CE period is the 11[th] coldest period between 500 and 2000 CE (Appendix G). Climate model simulations which incorporate estimates of the stratospheric sulfate aerosol forcing based on ice core records produce NH summer land temperature anomalies of around –0.3 °C, while individual ensemble members display cooling as large as –0.8°C, comparable to the tree ring-based estimates. The model simulations thus suggest that the tree-ring-derived cooling is explainable as a result of the combined effects of internal climate variability and volcanic aerosol forced cooling. The spatial patterns of the summer temperature decrease generally agree with the tree-ring-based reconstructions and the ensemble model simulations. For example, the growing season cooling registered in the NH tree-ring records is initially pronounced in western and central Europe and Scandinavia, with colder conditions in Alaska, which generally aligns with the spatial patterns of cooling found in the ensemble means of climate simulations. The peak summer cooling in the tree-ring records in 853 CE is influenced by a shift to cold conditions central Asia and Siberia. The cooling in these regions is also expressed in the model simulations in the summers of 853 and 854 CE, although with reduced amplitudes of temperature variability compared with the tree-ring temperature records. Strong cooling in central Asia and Siberia has been reconstructed from tree-rings in the years following many other large eruptions in the Common Era, such as the assumed Mount Asama eruption and unidentified volcanic eruptions in 1109 CE (Guillet et al., 2020), the Mount Samalas eruption in 1257 CE (Guillet et al., 2017), an unidentified eruption in 1453 CE (Stoffel et al., 2015; Abbott et al., 2021) and the Huaynaputina eruption in 1601 CE (White et al., in review).

In some respects, however, the spatial patterns differ between the climate model simulations and the tree-ring reconstructions. In particular, the persistent cool conditions in central Asia and Siberia in 855 CE are only found in the tree-ring-based reconstructions. These deviations (changes in temperature amplitudes and in spatial patterns) are



expected as the ensemble means of the simulations focus on the signal of the volcanic eruption by reducing internal
climate system variability. In contrast the tree-ring-based reconstruction contains both internal variability and the
potential forcing signal of the eruption and/or other external drivers. Therefore, the reconstructed cooling in Asia
and Siberia in 855 CE is potentially related to internal variability of the climate, such as changes in the large-scale
atmospheric circulation rather than being externally forced by the Churchill eruption.
The reconstructed climatic cooling peak in 853 CE aligns with the eruption date of the winter 852/3 CE Churchill
eruption but the timing of the start of this tree-ring-inferred cooling trend begins in summer of 851 CE, thereby
predating the eruption (and its associated age uncertainty). However, the magnitude of the temperature decline in
summer 851 CE is within the range of natural temperature variability and it is not until the summers of 852 and 853
CE when temperatures exceed the range of natural variability. The modelled climate scenario cooling occurs later in
853 CE, with widespread cooling present in summer 854 CE and winter 854 CE. The results from the tree-ring-
based temperature reconstructions preclude attribution of the climatic cooling solely to the Churchill eruption, but
the eruption timing clearly corresponds with cooling as registered in both reconstructed and simulated approaches.
These findings therefore suggest that the winter 852/3 CE Churchill eruption exacerbated a naturally occurring cold
period. This is supported by the decadal-scale step changes in temperatures recorded in the tree-ring-based
reconstructions (Fig. 4a) and NGRIP1 $\delta^{18}$O reconstructions (Fig. 6, Appendix J) prior to and after 852/853 CE.
Hydroclimate changes driven by volcanic eruptions are less clearly defined than those of temperature, partly due to
the higher degree of variability in precipitation and the small changes in atmospheric moisture associated with the
magnitude of temperature change often associated with volcanic-cooling. For example, the Clausius–Clapeyron
relationship predicts that the water-holding capacity of the atmosphere decreases by approximately 7% for every
1°C cooling (Held and Soden, 2006). Therefore, moisture changes associated with the 852/3 CE Churchill eruption
would be expected to be in the order of ca. <5%. Some observational and modelling studies have, however, reported
a reduction in global precipitation following explosive volcanic eruptions (e.g. Robock and Liu, 1994; Iles et al.,
2013). Evidence to support a change in precipitation driven by the 852/3 CE eruption is lacking: no statistical
changes were detected in NH precipitation variability during the 853 CE eruption period in this study and spatial
patterns of precipitation reconstructed from climate modelling and tree-rings are inconsistent, suggesting that
internal climate system variability dominates. There is also no evidence from the palaeoenvironmental
reconstructions to support hydrological changes on multidecadal time scales as changes in the peatland water depths
differ spatially and temporarily and most records present longer centennial-scale changes that do not correspond
with the eruption date.
The climate forcing of the 852/3 CE Churchill eruption derived from existing ice-core records and used in the
climate model simulations is the current best estimate. Uncertainty in the stratospheric aerosol forcing (as shown in
Fig 3b) is not incorporated into the model simulations as e.g., was done by Timmreck et al. (2021). Furthermore,
additional forcing factors have not been explicitly taken into account. In particular,  this explosive eruption is
characterised by high chlorine concentrations in the ice-cores (Fig. 2) and a very extensive ash-cloud across the NH



mid to high latitudes, suggesting large atmospheric loadings. Emissions of halogens and ash have the potential to
influence climate but their climate forcing potential is poorly constrained and so they remain unaccounted for in the
EVA and EVA_H forcing time series, as well as in the CESM simulations. The injection of a large quantity of
halogens along with sulfur by the 852/3 CE eruption may have modulated the impact on surface temperatures: some
model studies suggest coemission of halogens may intensify or prolong the volcanic cooling (Wade et al. 2020,
Staunton-Sykes et al. 2021), although contrasting model results suggest the effect may be model or event dependent
(Brenna et al. 2020). The influence of ash on radiative forcing is currently unclear. For example, recent observations
for the Kelud 2014 eruption suggest that ash exerted a radiative forcing of -0.08 W/m$^2$ three months after the
eruption (Vernier et al. 2016), even though the volcano erupted only $0.5 \pm 0.2$ x $10^{11}$ kg of ash (Maeno et al. 2019,
Aubry et al., 2021). In comparison, we found that the Churchill eruption erupted 4.9 x $10^{13}$ kg (3.9–6.1 x $10^{13}$ kg) of
ash, which might suggest a potentially strong radiative forcing from ash that is unaccounted for in our modelling.
However, the short lifetime of ash in the atmosphere makes it questionable whether the associated forcing would
persist long enough to significantly affect surface temperature and tree-ring growth. Furthermore, the co-injection of
ash with sulfur could likely reduce the radiative forcing associated with sulfate aerosol since ash particles uptake
sulfur dioxide, thereby reducing its lifetime (Zhu et al. 2020).
**4.2 Climatic-Societal impacts of the 852/3 CE Churchill eruption**
The White River Ash east (WRAe) deposit from the 852/3 CE Churchill eruption has reported thicknesses of 50–80
m proximal to Mount Churchill, and visibly extends in an easterly direction >1,300 km from the source (e.g.,
Richter et al., 1995; Lerbekmo 2008; Patterson et al., 2017). The considerable ash fallout synonymous with this
eruption had lasting environmental and societal consequences for regions proximal to the source, driven primarily
by the physical and chemical impacts of emissions from the eruption. Known impacts include changes in vegetation
and wetland ecology (e.g. Rainville, 2016; Payne and Blackford, 2008; Bunbury and Gajewski, 2013) and
displacement of local human populations (e.g. Kristensen et al., 2020; Hare et al., 2004; Mullen, 2012).
Historical records gleaned from a wide range of sources across Europe, Africa and Asia provide an opportunity to (i)
assess the extent to which the 852/3 CE Churchill eruption had distal societal consequences, (ii) corroborate or
critique results from the modelled and tree-ring-based climate scenarios around the time of the Churchill eruption,
and (iii) identify any evidence of extreme weather conditions that is not registered in the paleoenvironmental
reconstructions, such as severe winters. European historical records spanning the 850s document some anomalous
conditions, albeit fewer extreme weather events and associated crises than in other decades of the 9[th] century (Fig.
8). Food shortages and extreme weather were reported shortly before and after the 852/3 CE eruption in western
Germany; a severe subsistence crisis may have also occurred in nearby northern France and Belgium that set in
during the eruption year or shortly thereafter. Tree-ring reconstructions show that the growing season in 852 CE was
particularly cold in Europe, with temperature declines of –2°C or more in northwest Europe, and simulated
temperatures also show a temperature decline, albeit to a lesser magnitude (ca. –0.2°C). The cause of the crisis in
Germany in 853 CE is not detailed in the sources, nor are weather extremes observed that would corroborate the



inferred temperature anomaly. An extreme winter is identified as the cause of a food shortage reported in northern
France and Belgium in the early 850s, but the dates of the winter and the food shortage are not certain. It is notable
that a particularly sustained effort to record natural phenomena (including extreme weather) was undertaken in Irish
monasteries in the ninth and adjacent centuries (McCarthy and Breen, 1997; McCarthy, 2008), perhaps making it
more likely that unusual weather would be recorded here even in the absence of major societal impacts. Moreover, a
survey of 1,219 years of reporting of severe cold (mainly in winter) in Irish annals has revealed a repeated link to
explosive volcanism as registered in elevated Greenland sulphate, such that the medieval Irish may have been
particularly acute observers of volcanic winter-season impacts (Ludlow et al., 2013). However, there are no reported
extreme weather events in Ireland in the early 850s. Repeated reports of extreme cold occur from April of 855 CE to
winter 855/6 CE for Ireland occur amidst a return to average climatic conditions in the tree-ring reconstructions and
modelled climate scenarios for western Europe. Elsewhere, in the area of Huainan, China, drought is recorded in 852
CE, 854 CE and 855 CE and is consistent with the expected impacts of high-latitude Northern Hemispheric volcanic
eruptions on the East Asian summer monsoon (e.g., Zhuo et al, 2014; Iles and Hegerl, 2015). There is, therefore,
some agreement between the historical records and reconstructed and modelled climate, but not uniformly so
between 851-855 CE.
The pollen records are insufficiently resolved to identify sub-decadal anomalies or extreme weather events, but they
provide a useful longer-term perspective on societal adaptation to climate variability. Precise comparisons of the
pollen assemblages between sites are facilitated by the presence of WRAe, which dispels any chronological
uncertainty with respect to the timing of changes in land-use. The pollen records clearly show spatially complex
patterns in the extent and intensity of land-use, implying that changes in human activity around this time were not
driven merely by responses to changing environmental conditions. Rather, it would seem that any observed cultural
shifts around this time reflect an interplay of social, economic and political factors.

### 632    4.3 Transatlantic comparisons of terrestrial hydroclimate change in the Medieval Period

The MCA is commonly characterised as a warm period ca. 950–1250 CE (Mann et al., 2009), with dry conditions in
Europe and North America (e.g. Büntgen and Tegel, 2011; Ladd et al., 2018; Marlon et al., 2017). There is,
however, considerable spatial variability in the timings of the MCA onset and peak warmth (e.g. Neukom et al.,
2019) as well as hydroclimatic expressions (e.g. Shuman et al., 2018). In order to assess regional variability in
terrestrial MCA hydroclimate across northeastern North America and western Europe, this study provides
chronologically precise hydroclimatic comparisons facilitated by the detection of the WRAe isochron in our
peatland archives as well as the NGRIP1 ice core, which acts as a chronological tie point between the
palaeoenvironmental reconstructions. Comparisons of our eleven peatland records show that there is no consistent
multidecadal-scale hydrological response associated with the MCA; rather hydrological conditions are variable both
within and between records. There are also no clear temperature trends associated with the MCA detected in the
NGRIP1 $\delta^{18}$O record (Fig. 6): temperatures are elevated in central Greenland during the 10$^{th}$ century but these are
not sustained during the remainder of the medieval period, which is generally characterised by cooler, fluctuating

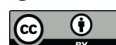

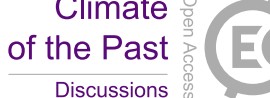

temperatures. These findings suggest that there is no clear climatic expression of the MCA in the North Atlantic
region.
The environmental reconstructions presented in this study highlight the heterogeneous and time-transgressive nature
of the reconstructed MCA hydroclimate change. For example, a dry shift, which may be typical of a MCA climate
response, began ca. 900 CE in northern Nova Scotia and some records in Newfoundland, and corresponds with a
change to warmer conditions in central Greenland. However, the onset of drier conditions is delayed by ca. 100
years in more south-westerly sites in Nova Scotia and Maine as well as on the east coast of the North Atlantic in
Ireland. In addition, all peatland records contain temporary wet shifts that occur prior to the MCA ca. 700-850 CE,
which corresponds with a period of generally colder temperatures in central Greenland as reconstructed from the
NGRIP1 $\delta^{18}$O. The timings and extent of the wet shifts vary, however, between peatland records with no clear
spatial patterns to provide insight into the climate forcing mechanism driving this change. NGRIP1 records another
more abrupt and pronounced temperature decrease ca. 1000–1050 CE, the time of which corresponds to a temporary
wet shift in several peatland records from Ireland, Newfoundland, Nova Scotia and Maine. However, once again the
timing and extent of these wet shifts vary between reconstructions. The chronological precision afforded by the
presence of the WRAe isochron in climate reconstructed presented in this study therefore conclusively demonstrates
that the differences in the timings of hydroclimatic change between records reflect a true difference in peatland
responses to environmental conditions and are not a feature of chronological uncertainty generated from the age-
depth modelling process.
Here we have reported the dominant peatland hydroclimatic patterns that are supported by multiple regional
peatland records; however, some differences exist between proximal reconstructions, such as the clusters of three
peatland records developed within ca. 10 km in eastern Newfoundland and two records within ca. 110 km in Maine.
The differences in hydroclimate at such local levels in Newfoundland may reflect the degree of spatial hydroclimate
variability during this period, but also may be exacerbated by autogenic-driven peatland responses such as enhanced
peat accumulation under warmer MCA climates that would drive an apparent lowering of the water table (e.g.
Swindles et al., 2012). The divergence between the hydroclimate reconstructions obtained from the Maine peatlands
is likely influenced by a fire disturbance event at one of the sites, Saco Heath, which created a substantial hiatus in
peat accumulation in some areas of the site (Clifford and Booth, 2013). Whilst the Saco Heath record presented here
appears less impacted by the fire, there is a high degree uncertainty in the hydroclimate reconstruction between ca.
1000-1250 CE when the accumulation rate slows, which may reflect a temporary hiatus (Figure 6; Appendix D).
The development of more palaeoenvironmental reconstructions from sites containing the WRAe, particularly in
locations such as Maine and western Europe, will be useful to investigate further MCA trends further.
**5. Conclusions**
The winter 852/3 ± 1 CE Churchill eruption was one of the largest magnitude volcanic events of the first
millennium. Tree-ring temperature reconstructions show a NH summer temperature anomaly of around –0.8°C in



853 CE, and the corresponding 3-year mean temperature anomaly ranks as the 11[th] coldest over the 500-2000 CE
period (Appendix G). On the other hand, the reconstructed climate forcing potential (i.e. atmospheric sulfate burden)
of this eruption derived from ice core records is moderate, smaller than that associated with the 1991 Pinatubo
eruption. This apparent mismatch between forcing and response is, we find, explainable as resulting from the
combined impact of natural climate variability and volcanic aerosol forcing. Climate model simulations driven by
reconstructed aerosol forcing show an ensemble mean response of –0.3 $^{\circ}$C, but individual ensemble members that
show cooling of up to –0.8$^{\circ}$C comparable to the tree-ring reconstructions. Support for the correspondence of the
eruption with a naturally occurring cool period is provided by the timings of the cooling trend reconstructed by the
tree-rings, which begins in summer 851 CE and therefore predates the winter 851/2 to winter 853/4 CE age
uncertainty of the eruption, and the seasonal-scale NGRIP1 temperature reconstruction (Appendix J). The simulated
temperature response of the eruption may also be underestimated, because the forcing potential models do not
account for the potential role of halogens or volcanic ash, both of which show high atmospheric abundances after the
eruption. Further research in combined sulphur, halogen and ash modelling and better ice-core constraints about
their atmospheric loadings are therefore required to provide more holistic understandings of potential ash-rich
volcanic impacts on climate and society.
Areas proximal to Mount Churchill experienced widespread and prolonged ecological, environmental and societal
changes attributed to the eruption emissions, but there is no evidence of multidecadal-scale climatic response
preserved in distal palaeohydrological records from the North Atlantic region that are precisely temporally linked by
the 853 CE Churchill WRAe isochron. Pollen records of vegetation change and human activity from Ireland linked
by the WRAe isochron also provide no evidence to support long-lasting societal responses in Ireland associated with
the eruption. Evidence of short-term societal impacts in Europe from the 852/3 CE Churchill eruption remains
equivocal: some historical records from Ireland and Germany, and possibly northern France and Belgium, report
harsh winter conditions and food shortages within the age uncertainties of the eruption but similar events were
reported outside of the eruption period and were not unknown in the 9[th] century. The 852/3 CE Churchill eruption
therefore exemplifies the difficulties of identifying and confirming volcanic impacts on society even when only a
small eruption age uncertainty exists.
The presence of the WRAe isochron in peatlands in northeastern North America and western Europe assists with
comparisons of hydroclimatic reconstructions during the Medieval Climatic Anomaly, often defined as a period of
globally increased temperatures between 950–1250 CE (Mann et al., 2009). Reconstructed hydroclimate conditions
in 853 CE vary, highlighting leads and lags in the terrestrial responses to environmental change that may otherwise
be considered contemporaneous without the temporal precision provided by the WRAe. This study shows a lack of a
consistent terrestrial response to MCA climate change in the North Atlantic region; rather the MCA time period is
characterised by time-transgressive and heterogenous hydroclimatic conditions. These findings contribute to a
growing body of research that cautions against the application of the globally defined MCA characteristics when
interpreting individual records of palaeoenvironmental change and ultimately questions the detectability of a
coherent MCA climate signature.



**Appendices**

**Appendix A: Additional methodological information to support the forcing potential reconstructions (Section 2.2)**

The EVA (eVolv2k) reconstruction (Toohey and Sigl, 2017) is the recommended volcanic forcing dataset for climate model simulations of the Paleoclimate Modeling Intercomparison Project (PMIP, Jungclaus et al., 2017; Kageyama et al., 2018). The EVA reconstruction uses volcanic stratospheric sulfur injection estimates derived from sulfate deposition from an extensive bipolar array of ice cores (Sigl et al. 2015), which are then converted into an SAOD time series using the idealized, scaling based aerosol model Easy Volcanic Aerosol (EVA, Toohey et al., 2016). The global mean radiative forcing (RF) time series is estimated from the SAOD using the following relationship from Marshall et al. (2020):

$$RF = -19.2 \times (1-e^{-SAOD}) \tag{1}$$

where RF is in W m$^{-2}$.

**Appendix B: Additional methodological information to support the climate model simulations (Section 2.3)**

An initial condition ensemble simulation was created using the Community Earth System Model version 1.2.2 (CESM), consisting of 20 ensemble members. CESM is a state-of-the-art fully coupled Earth system model composed of atmosphere, land, ocean, and sea ice components. To generate the ensemble members, initially a seamless transient simulation was run from 1501 BCE (Kim et al., 2021) with time-varying orbital parameters (Berger, 1978), TSI (Vieira et al., 2011; Usoskin et al., 2014, 2016), GHG (Joos and Spahni, 2008; Bereiter et al., 2015), and volcanic forcing from the HolVol v.1.0 (Sigl et al., 2021) and eVolv2k (Toohey and Sigl, 2017) databases. The necessary prescribed spatial-temporal distribution of volcanic sulfate aerosol for the simulation is generated using the Easy Volcanic Aerosol Model (EVA, Toohey et al., 2016) and following the same procedure employed by McConnell et al. (2020) and Kim et al. (2021). In the procedure, the EVA-generated spatio-temporal distribution of sulfate was first converted to volcanic aerosol mass to be readable by CESM. This distribution of volcanic aerosol mass in CESM was scaled up by 1.49 to reconcile CESM and EVA atmospheric responses to the 1991 Pinatubo eruption. Then, the transient simulation was branched off at 845 CE and a small perturbation was introduced at the first time step in the atmosphere. The 20 ensemble members were run from this point until 859 CE. During this 14 year period, no other volcanic eruptions were included except the 852/853 CE Churchill eruption.

Mann-Whitney U-test was used to test the statistical significance of changes in temperature and precipitation after the Churchill eruption. The null hypothesis of a Mann-Whitney U-test test states that the two datasets share the same statistical distribution derived from the same population. In this study, the distributions of the temperature and precipitation anomalies after Churchill eruption (853, 854 and 855 CE individually) derived from all ensemble members were compared to those of the 845–852 CE pre-eruption period ensemble. We assume that changes in temperature and precipitation after the eruption are statistically significant if the null hypothesis of the Mann-Whitney U-test test is rejected at 5% confidence level.



**Appendix C: Additional methodological information to support the NH tree-ring summer temperature**
**reconstructions (Section 2.4)**
We employed a principal component regression (PCR) to reconstruct NH JJA temperature anomalies (with respect to
1961–1990) from tree-ring records. We coupled this PCR with a bootstrap random sampling approach to quantify the
robustness of our reconstruction and to estimate confidence intervals of reconstructed JJA temperatures. To account
for the decreasing number of records available back in time, we combined the PCR with a nested approach. In total,
our reconstruction is based on 23 subsets of tree-ring chronologies or nests. The earliest and most recent nests span
the periods 500–551 and 1992–2000 CE, respectively. The most replicated nest (1230–1972 CE) includes 25
chronologies. For each nest, we reduced the proxy predictors matrix to principal components (PCs) using a Principal
Component Analysis (PCA). PCs with eigenvalues >1 were included as predictors in multiple linear regression models
calibrated on JJA temperature (1805–1972 CE) extracted from the Berkeley Earth Surface (BEST) dataset
(http://berkeleyearth.org/data/). We assessed the robustness of each model using a split calibration–verification
procedure using a bootstrap approach repeated 1,000 times. We computed the final reconstruction of each nest as the
median of the 1,000 realizations. The final 500–2000 CE reconstruction combines the 23 nests with their mean and
variance adjusted to be identical to the 1230–1972 CE most replicated one. To place the summer temperature
anomalies within the context of climate variability at the time of major volcanic eruptions, we removed longer
timescale variations by filtering the final reconstruction, which involved calculating the difference between the raw
time series and the 31-yr running mean.
The target field (predictand) used for the reconstruction is the BEST JJA gridded temperature dataset (1°×1° latitude-
longitude grid). We divided the NH into 11 subregions defined according to the spatial distribution of the 25 tree-ring
records and their correlation. Chronologies were grouped in the same subregion when their correlation coefficients
over their overlapping period exceeded 0.3. Only one chronology was included in the Quebec, Western and Central
Europe, Siberia - Taymir, Siberia - Yakutia, and China - Qilian Mountains subregions. In these clusters, we used an
ordinary least square regression to reconstruct JJA temperatures. In the other subregions such as Western and Central
Europe – which includes five TRW and MXD records – we used the nested PCR approach (see above) to reconstruct
gridded summer temperature anomalies. Based on this approach, we reconstructed robust temperature anomalies back
to 500 CE for 3486 NH grid points.





**Appendix D: Peatland chronologies**









**Fig. D1: (a-o): Core chronologies (Sup. Fig. 1a-o) were developed using Bayesian analysis within the R package "BACON" (Blaauw and Christen, 2011) based on ¹⁴C dates (calibrated using NH IntCal20 calibration Curve (Reimer et al., 2020)) and tephrochronologies. (p): Shard counts and major-minor element glass compositions for the WRAe in Southwest Pond Bog. Comparative glass electron probe microanalysis data (UA1119) is taken from Jensen et al. (2014).**



**Appendix E: Composite testate amoebae-inferred peatland water table record from Sidney Bog, Maine**

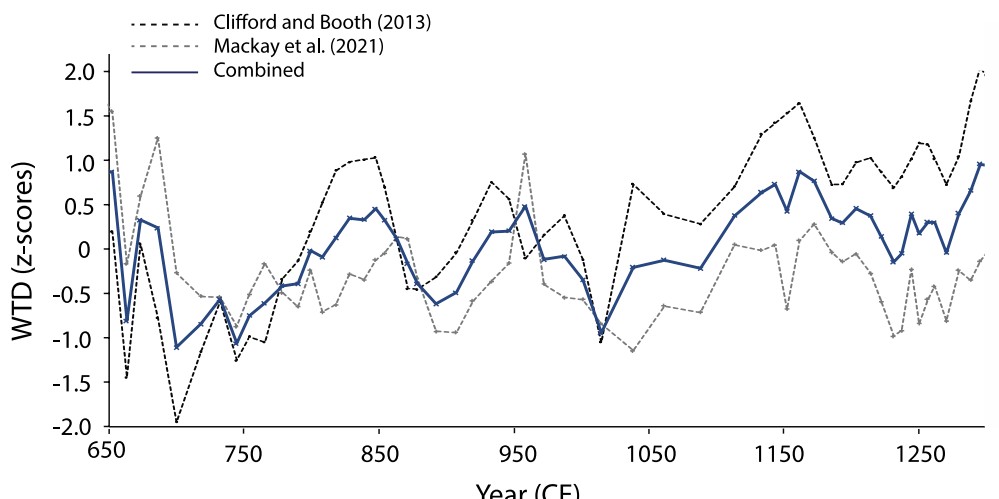


**Fig. E1: Composite WTD record for Sidney Bog, Maine, USA based on testate amoebae assemblage data obtained from**
**two cores obtained from different coring location on the peatland (Clifford and Booth, 2013; Mackay et al., 2021). Testate**
**amoebae water table depth (WTD) reconstructions were obtained using the tolerance-downweighted weighted averaging**
**model with inverse deshrinking (WA-Tol inv) from the North American transfer function of Amesbury et al. (2018). To**
**produce the composite record, the chronological resolution of the Clifford and Booth (2013) WTD record has been**
**increased to the same resolution as the Mackay et al. (2021) record using linear interpolation between chronological**
**adjacent WTD values. The composite record then presents the average WTD of the interpolated Clifford and Booth**
**(2013) and the Mackay et al. (2021) reconstructions.**



**Appendix F: Tree-ring inferred NH summer temperature anomalies**

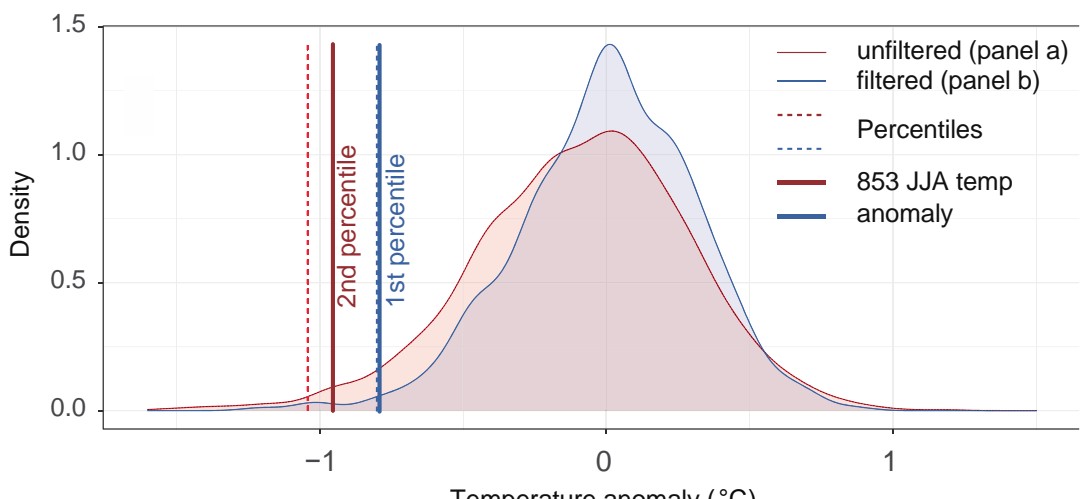

**Fig. F1: Distributions of JJA temp. anomalies in the unfiltered (blue) and filtered (red, 31 yr-mov. av. Filter) 500-2000 CE,**
**NH reconstructions. Blue and red vertical dotted bars indicate the 1ˢᵗ percentile of the filtered and the 2ⁿᵈ percentile of the**
**unfiltered reconstructions, respectively. Blue and red vertical lines show the cooling observed in 853 CE in the filtered and**
**unfiltered reconstructions, respectively.**





**Appendix G: Eruption information for tree-ring inferred coldest years between 500–2000 CE**

Table G1: Top 30 coldest years during the period of 500–2000 CE based on tree-ring temperature anomalies filtered using a 3-year mean and corresponding eruption information for proximal calendar years. Eruption dates and volcanic stratospheric sulfate injection (VSSI) estimates taken from the eVolv2k reconstruction (Toohey and Sigl, 2017), representing the most immediate preceding eruption in the data set. Black = eruption occurred within 2 years of the coldest reported year; grey = eruption occurred more than 2 years before the coldest reported year.

| Rank | Year (CE) | Temperature anomaly | Preceding eruption | VSSI | Time difference (eruption year – cold year) |
|---|---|---|---|---|---|
| 1 | 536 | -1.40 | 536 UE | 18.8 | 0 |
| 2 | 627 | -1.25 | 626 UE | 13.2 | -1 |
| 3 | 1601 | -1.25 | Huyaniputina 1600 | 19.0 | -1 |
| 4 | 1783 | -1.21 | Laki 1783 | 20.8 | 0 |
| 5 | 1453 | -1.09 | 1453 UE | 10.0 | 0 |
| 6 | 1109 | -1.02 | 1108 UE | 19.2 | -1 |
| 7 | 1032 | -0.95 | 1028 UE | 7.8 | -4 |
| 8 | 1259 | -0.86 | Samalas 1257 | 59.4 | -2 |
| 9 | 800 | -0.81 | 800 UE | 2.5 | 0 |
| 10 | 1463 | -0.74 | 1458 UE | 33.0 | -5 |
| 11 | **853** | **-0.71** | **Churchill 852/853** | **2.5** | **-1/0** |
| 12 | 1816 | -0.71 | Tambora 1815 | 28.1 | -1 |
| 13 | 979 | -0.69 | 976 UE | 6.2 | -3 |
| 14 | 1833 | -0.69 | 1831 Babuyan | 13.0 | -2 |
| 15 | 1589 | -0.65 | Colima 1585 | 8.5 | -4 |
| 16 | 1699 | -0.64 | 1695 UE | 15.7 | -4 |
| 17 | 1641 | -0.62 | 1640 Parker | 18.7 | -1 |
| 18 | 637 | -0.57 | 637 UE | 1.7 | 0 |
| 19 | 903 | -0.54 | 900 UE | 5.6 | -3 |
| 20 | 1459 | -0.53 | 1458 UE | 33.0 | -1 |
| 21 | 1677 | -0.52 | 1673 UE | 4.7 | -4 |
| 22 | 1697 | -0.44 | 1695 UE | 15.7 | -2 |
| 23 | 639 | -0.35 | 637 UE | 1.7 | -2 |
| 24 | 541 | -0.34 | 540 UE | 31.9 | -1 |
| 25 | 543 | -0.32 | 540 UE | 31.9 | -3 |
| 26 | 1835 | -0.31 | Cosiguina 1835 | 9.5 | 0 |
| 27 | 1643 | -0.29 | 1640 Parker | 18.7 | -3 |
| 28 | 546 | -0.26 | 540 UE | 31.9 | -6 |
| 29 | 538 | -0.25 | 536 UE | 18.8 | -2 |
| 30 | 640 | -0.07 | 637 UE | 1.7 | -3 |






**Appendix H: Climate model simulations of NH summer and winter precipitation anomalies between 856-858**
**CE**

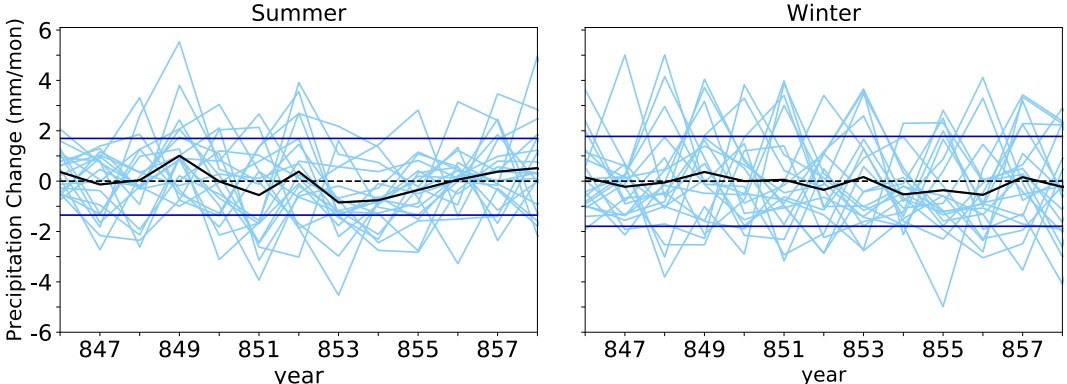

**Fig. H1: The spatially-averaged NH extratropical (15° – 90°N latitudes) precipitation anomalies from 20 ensemble**
**simulations for summer (JJA) and winter (DJF) in light blue lines. The thick black lines indicate the ensemble means and**
**the horizontal blue lines represent one standard deviation from the ensemble means of the 845 – 852 CE pre-eruption**
**period.**
**Appendix J: NGRIP1 δ¹⁸O isotopes temperature reconstruction (9$^{th}$ century)**

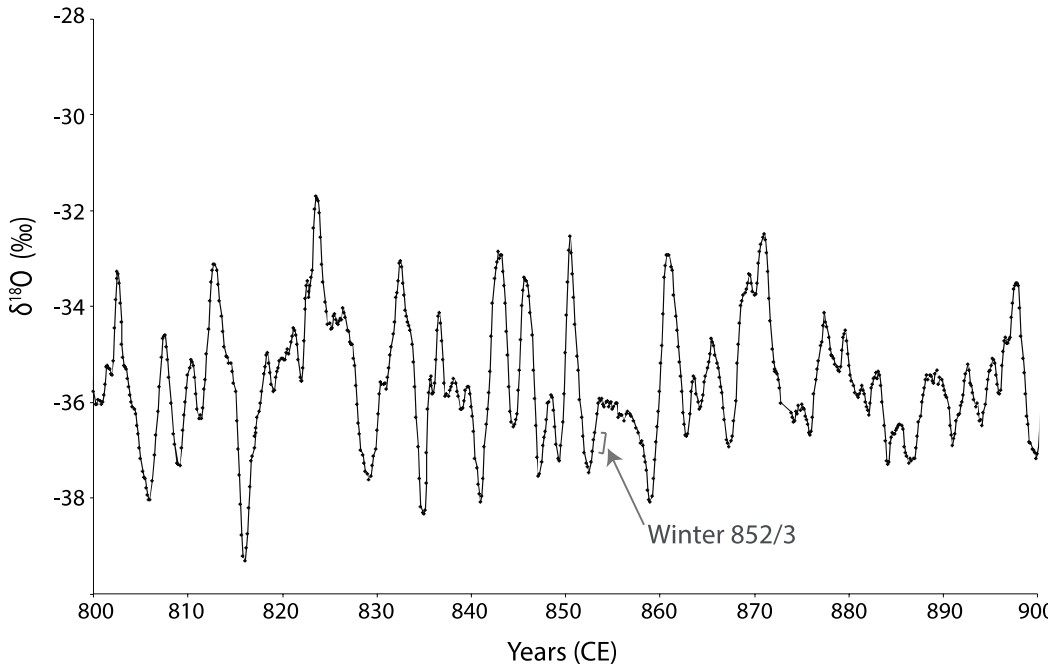


**Fig. J1: NGRIP1 δ¹⁸O isotopes temperature reconstruction (Vinther et al., 2006), plotted on NS1-2011 chronology (Sigl et**
**al., 2015). Warmer (colder) temperatures are represented by higher (lower) δ ¹⁸O values. The eruption age estimate for the**
**852/3 CE Churchill eruption is denoted.**



## Author contributions

HM, GP and BJ were responsible for the conceptualization and design of the project. HM, MA, AM, AB and GS conducted the testate amoebae analyses as well as the associated data analysis and interpretation. HM and MA conducted the testate amoebae analyses as part of projects supervised by PDMH, PGL and DC. RB and HM created the Sidney Bog composite testate amoebae record. GP and LCM designed and conducted the Irish pollen and tephra analysis for the Irish sites (exception of Dead Island record, tephrochronology by GS). TA and MT designed the forcing potential analyses, which were conducted by TA. MSigl analyses the ice-core chronologies and associated data. BJ and MB designed the eruption volume estimate and magnitude analyses, which were conducted by MB. WK and CR designed the climate model simulation analyses, which were conducted by WK. CC and MStoffel designed the tree-ring temperature reconstruction analyses, which were conducted by CC. KJA designed and analysed the tree-ring drought reconstructions. JM, TPN, NDC, FL, CK and ZY analysed the historical records. HM, KLD, TA, WK, CC, AM, KA and MB designed and produced the visualisations. HM prepared the original draft of the manuscript and all co-authors were involved in the writing review and editing process.

## Competing Interests

The authors declare that they have no conflict of interest.

## Acknowledgements

This paper benefitted from discussion at events of the Past Global Changes (PAGES) working group 'Volcanic Impacts on Climate and Society' (VICS), as well as with Angus M. Duncan and Richard J. Payne. PAGES is supported by the Chinese Academy of Sciences (CAS), and Swiss Academy of Sciences (SCNAT). H. Mackay and M. Amesbury were supported by the UK Natural Environment Research Council (PRECIP project grants NE/G019851/1, NE/G020272/1, NE/G019673/1 and NE/G02006X/1 and MILLIPEAT project grant NE/1012915/1). A Quaternary Research Association New Research Workers award granted to H. Mackay and the NERC Radiocarbon Facility NRCF010001 (allocation numbers 1744.1013 and 1789.0414). C. Corona and M. Stoffel were supported by the Swiss National Science Foundation Sinergia project CALDERA (grant agreement no. CRSII5_183571). WMK and CCR are supported by the Swiss National Science Foundation (SNSF, grants 200020_172745 and 200020_200492). The climate mode simulations were performed at the Swiss National Super Computing Centre (CSCS). M. Sigl acknowledges funding from the European Research Council (ERC) under the European Union's Horizon 2020 research and innovation program (grant agreement 820047). F. Ludlow and C. Kostick were supported by an Irish Research Council Laureate Award (CLICAB, IRCLA/2017/303). J.G. Manning and F. Ludlow also acknowledge support from US National Science Foundation Award # 1824770.



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
