# Peer review of "The 852/3 CE Mount Churchill eruption: examining the potential 1"

_Climate of the Past, 2021_

## Author Response (AR1)

**Mackay et al. Mount Churchill Climate of the Past**

**Response to reviewer comments**

CC1

I read with great pleasure the manuscript "The 852/3 CE Mount Churchill eruption: examining the potential climatic and societal impacts and the timing of the Medieval Climate Anomaly in the North Atlantic Region" by Helen Mackay and co-authors. I think the journal Climate of the Past is the it home for this manuscript. I found this work to be meaningful and comprehensive research, including many details, although each detail seems to be necessary and does not overburden the investigation. To my mind, the manuscript represents a synthesis of diverse facts from different fields of science, which focus on the 852/3 CE eruption event and taking together help to sort out the influence of this event on palaeoenvironment and social life of mankind. In reality, this is a very difficult task to follow (reconstruct) the impact of a single volcanic eruption on the environment using a series of diverse palaeoecological records (peat, lake sediments, ice and so on) accompanied with surviving historical documents. I think this research is a good example of testing the potential of complex paleoenvironmental reconstructions to date. It's not our fault that we can't reconstruct every detail we would like to do. The research is well organized, including a good selection of literature sources. Conclusions are plausible and discreet. I believe this manuscript deserves to be published without changes in this journal. It is actual and attractive. I only found a few misprints across the text of the manuscript and I propose to correct them. See below:

Line 8 – Pete G. Langdon. I think it is Peter (not Pete).

Line 106 – Sigl et al. 2015 – the comma is missing; Fig 2 – the point is missing.

Line 219 – (Fig. d-e) – the number of Fig. is missing. I think this is 1.

Line 258 – (Newfield 2013, Devroey 2019) – two commas are missing and one comma instead of semicolon.

Line 572 – (Fig 3b) – the point is missing.

RESPONSE: We thank the reviewer for their comments and for very helpfully reporting the misprints. We have now incorporated all suggested changes in text, with the exception of changing 'Pete' to 'Peter' on Line 8.

January 19, 2022

Dear Dr. LeGrande,

I have completed reviewing the manuscript "The 852/3 CE Mount Churchill eruption: examining the potential climatic and societal impacts and the timing of the Medieval Climate Anomaly in the North Atlantic Region" by Mackay et al. The 852/3 Churchill eruption was one of the largest eruptions in the first millennium in terms of its explosivity (but not aerosol loading). 853 CE also consistently shows up as an unusually cold year in northern hemisphere temperature reconstructions using tree-rings. In their study the authors first examine the timing of the eruption using two Greenland ice-cores and conclude that the eruption likely occurred in the winter of 852/853. They then reconstruct the stratospheric aerosol optical depth for the eruption. Next, they compare tree-ring reconstructed and climate model simulated post-eruptive cooling and hydroclimate change over parts of the Northern Hemisphere. The find that climate model simulated cooling is lower than tree-ring reconstructed cooling. Further, the authors use North American and European peat cores and find no consistent low-frequency hydroclimate change post-eruption leading into the Medival Climate Anomaly period. Finally, the authors highlight the difficulties in attributing any historical evidence of subsistence crisis to the Churchill eruption. The manuscript is well-written, scientifically sound, extremely thorough, contributes significantly to our understanding of the Churchill eruption. It is also well-suited for publication in the the Climate of the Past Volcanic Impacts to Climate and Society Special Issue. Therefore, in my opinion I cannot offer much advice to the authors and do not have any requests to make in terms of revisions. I look forward to seeing the manuscript published soon.

Sincerely,

Mukund Palat Rao

REPONSE: We thank the reviewer for their positive comments and support for the manuscript.
Yours sincerely,
Helen Mackay (on behalf of all authors)

**Author Response to RC2**

- We thank the reviewer for their helpful and detailed comments, which have helped to strengthen the manuscript. We have provided a response to each comment below and stated the corresponding changes that have been made to the manuscript.

**Specific questions**

A main result is that the tree-ring reconstructed temperature suggests a significant cooling but the reconstructed sulfur emission and hence 'climate forcing potential' is suggested to be moderate, and the simulated cooling is smaller.

However, the authors only run climate model simulations based on SAOD reconstructions using the best-estimate of 5 Tg $SO_2$. Although it is briefly discussed that the climate forcing could have been underestimated due to missed interactions with halogens or ash, I think it would be useful to also include more discussion on the sulfur emission uncertainty. The $SO_2$ emission uncertainty is included for the EVA_H reconstruction in Figure 3, but not EVA – what is the peak SAOD if EVA is run using 7.5 Tg $SO_2$? This might be a useful addition to Figure 3. Is a higher EVA SAOD discounted because the EVA_H SAOD estimate is much lower? Given the limited ice core records could the emission estimate have been even higher? Some discussion on the spatial SAOD pattern would also be useful - would these results be different if an aerosol-microphysical model was used such as in Toohey et al. (2019) who show strong confinement of the aerosol to the NH for an eruption at 56°N. Would this lead to a stronger NH forcing and temperature response?

- RESPONSE: We have now included the 95% confidence interval on the EVA global mean SAOD prediction in Figure 3.b and quoted the corresponding number in the text (Line 268-270):

  "The EVA(eVolv2k) reconstructed stratospheric aerosol optical depth (SAOD) at 550 nm for the 852/3 CE eruption is relatively moderate, with a peak aerosol optical depth perturbation of 0.049 (95% confidence interval 0.021–0.085) in terms of global monthly mean, and 0.078 in terms of NH monthly mean (Fig. 3a-b)."

- We have also corrected a mistake: the uncertainty on the injected $SO_2$ mass is 3.3 Tg of $SO_2$ (Line 166), not 2.5 as wrongly reported in the initial submission. To further discuss uncertainty, we provide lower and upper-end estimate of the peak global mean radiative forcing (-1.7- -0.33W.m$^{-2}$) (Lines 281-284) based on the upper and lower end SAOD estimates as well as the range of estimates for the forcing efficiency depending on the eruption season (Marshall et al., 2019):

  "The upper-end SAOD estimate from EVA(eVolv2k), obtained from a winter eruption (which would maximize the forcing efficiency, Marshall et al., 2019), has a global monthly mean radiative forcing peak of -1.7 W m$^{-2}$. Conversely, the lower-end SAOD estimate from EVA_H, obtained from a summer eruption, has a mean peak forcing of -0.33 W m$^{-2}$."

  Furthermore, we now present in greater detail these uncertainties, and those related to the dispersion of the aerosol cloud, on Lines 543-549 (please see the quoted text below), and we discuss how they would affect the consistency between the simulated NH summer cooling and that reconstructed from tree rings. Although we cannot exclude that the mass of $SO_2$ emitted could be even higher than the upper-end used, we note that it is not necessary to

invoke such arguments as our simulations show that the combination of the forced response and natural variability could lead to the reconstructed NH cooling.

"Another possibility, which would relax the requirement for a rather strong contribution of natural variability, would be that the volcanic aerosol forcing was in reality stronger than that used here. With the upper-end reconstructed EVA(eVolv2k) SAOD estimate being 66% higher than the best estimate used in our model simulations, the forced model response could be even higher (Figure 3.b). Furthermore, a stronger restriction of aerosols to the NH, not simulated in the simple SAOD reconstruction methods but compatible with interactive stratospheric aerosol model simulations (e.g., Toohey et al., 2019) may also contribute to stronger aerosol forcing over the NH than used here."

L69 – I suggest changing 'precisely' to 'even when the eruption has a small age uncertainty' or similar

- RESPONSE: Change incorporated: wording changed as suggested above (now Line 68)

L124 – 1991 Mt. Pinatubo estimates are now slightly lower; perhaps say 'around a third'?

- RESPONSE: Change incorporated: wording changed as suggested above (now Line 121-122)

L173 and Appendix B – this description is a bit confusing - how are the ensemble members generated? Do they consider different ENSO states or is a small perturbation introduced for each? Could background conditions play a role in reconciling the model results and tree-ring reconstructions?

- RESPONSE: The ensemble members are generated by introducing a small perturbation for each simulation starting from the year 845 CE. We added a sentence clarifying the description in lines 177-188:

  "Each ensemble member is branched off at 845 CE with a small perturbation in the atmosphere introduced at the first time step. Then, the simulations are seamlessly run until 859 CE. "

  All simulations initially start from the same background conditions, but by the year of the eruption (853 CE), the range of ENSO states is diverse among the simulations as the SST is not fixed. Hence, ensembles members are considering different ENSO and initial conditions. Background conditions, not only from the ENSO but associated with internal variability, may play a role in reconciling the model and proxies. We have briefly mentioned the observed model-proxy discrepancy due to the internal variability in Lines 563-665.

  " These deviations (changes in temperature amplitudes and in spatial patterns) are expected as the ensemble means of the simulations focus on the signal of the volcanic eruption by reducing internal climate system variability."

Why does the model prescribe sulfate aerosol mass rather than the optical properties from EVA? This seems inconsistent with the focus on reconstructing the forcing up to this point. How does the model treat the aerosol-radiation interactions and what does the model then simulate for SAOD and radiative forcing and how does this compare to Figure 3?

- RESPONSE: CESM1.2 uses a prescribed monthly mean sulfate aerosol mass on a predefined latitudinal and vertical grid as an input volcanic forcing. The aerosol is then used in the

radiation code of the model, i.e., optical properties are estimated within the model assuming that the aerosol mass is comprised of 75 % sulfuric acid and 25% water and has a constant log-normal size distribution with a constant effective radius and following Neely et al. (2016). We included these details in Appendix B, Lines 771-775.

L183 – I'm a bit confused regarding the anomalies - are the volcanic anomalies with respect to the 845-859 period or pre-eruption (as specified for the statistical significance)? Please could you clarify. How sensitive are the anomalies to this reference period vs. for example the 5 years pre-eruption? Also for the tree-ring reconstructions – are the volcanic anomalies robust if a different reference period is taken such as the preceding 5 years?

- RESPONSE: The anomalies are calculated with respect to the 845-859 CE period, which is the entire simulated period (Lines 183-184). The statistical tests are, however, performed with respect to the pre-volcanic period 845-852 CE (Lines 188-191). The reason is that we would like to identify the strength of the changes in temperature and precipitation caused by the Churchill eruption compared to the eruption-free period, which is the period 845-852 CE.

Using a different period, for instance the pre-eruption period (845-852 CE), increases slightly the mean of the reference period, but the difference between global mean is rather minimal (See figure R1 below). Note that the analysis between the pre- and post-eruption temperature and precipitation is not affected by the choice of the period calculating the anomalies.

[Figure]

Fig R1. (left) Globally averaged multi-year monthly mean temperatures of two references periods: 845-852 CE (blue) and 845-859 CE (red), (right) differences between the monthly means of 845-852 CE and 845-859 CE.

L278 – It would be useful to have the radiative forcing simulated by CESM for comparison.

- RESPONSE: To diagnose the effective radiative forcing (ERF) from CESM, we would need to run simulations with fixed SSTs, which is outside the scope of this work. For this reason we still only present the scaling-based ERF estimates in our revised manuscript.

Fig. 4 – Please introduce NVOLC in the main text. The ylabels are also inconsistent (change vs anomaly). A legend on panels (c) and (d) would also be useful.

- RESPONSE: Changes are incorporated: we have introduced NVOLC to the main text (now Line 197). The legend for panels (c) and (d) is now included and the ylabels have been standardised.

Fig. 5 – Please specify that (b) (c) (e) and (f) are simulated anomalies. Why is (d) in a different projection?

- RESPONSE: Changes incorporated: "simulated" anomalies is highlighted in the titles and captions for Fig. 5 (b), (c), (e) and (f). The projection of (d) has been altered to match the other projections in this figure.

L521 – consider rephrasing to 'climate model simulations run with/using estimates of the stratospheric aerosol'

- REPONSE: change incorporated: rephrased to "climate model simulations using estimates of the stratospheric sulfate aerosol.." as suggested above (now Line 539).

L558 – 561: I think it would be useful here to also briefly discuss dynamically driven precipitation changes

- RESPONSE: Change incorporated: we have included a brief discussion in Lines 582-595 (it is brief to reflect the reviewers comment and because we do not see a consistent response in precipitation).

  "In principle, two possible processes might lead to precipitation changes after an eruption: thermodynamic or dynamic affects. The direct thermodynamics effect is related to the Clausius–Clapeyron relationship, which predicts that the water-holding capacity of the atmosphere decreases by approximately 7% for every 1°C cooling (Held and Soden, 2006). Therefore, moisture changes associated with the 852/3 CE Churchill eruption would be expected to be in the order of ca. <5%. Some observational and modelling studies have, however, reported a reduction in global precipitation following explosive volcanic eruptions (e.g. Robock and Liu, 1994; Iles and Hegerl, 2014, 2015). Beyond the thermodynamic affects, volcanic eruptions may also generate hydroclimate anomalies through changes in large-scale ocean-atmosphere circulation, including shifts in the latitudinal position of the Intertropical Convergence Zone (ITCZ; Haywood et al., 2013; Colose et al., 2016), an anomalously positive NAO or Northern Annular Mode (e.g. Christiansen, 2008; Stenchikov et al., 2006; Raible et al., 2016), a poleward jet shift (Barnes et al., 2016), and/or a narrowing of the Hadley Circulation (Ménégoz et al., 2018). The potential hydroclimate response to a high latitude eruption is therefore complex, reflecting the multiple, combined, and interacting effects of direct radiative forcing, feedbacks in those through changes in ocean-atmosphere circulation, and internal stochastic variability."

L725 – consider also adding the relationship for winter extratropical eruptions, which gives a slightly higher forcing of -1 Wm$^{-2}$. More importantly, what does CESM simulate? Is the RF different to that based on this scaling?

- RESPONSE: We now provide the range of forcing efficiency in Appendix A (Lines 763-764):

  ". The scaling pre-factor may vary between -20.9 and -17.4 W m$^{-2}$ depending on the eruption season."

Furthermore, we include the full range of possible forcing (Lines 281-284), please refer to our response to the first comment) accounting for the uncertainty on both SAOD and on forcing efficiency due to season of eruption. Please see our reply to your comment on L278 above for the CESM ERF response.

L738 - Where does the scaling of 1.49 come from? Why not run the model with the prescribed optical properties?

- RESPONSE: We followed a similar procedure used by Zhong et al. (2018) to create CESM-mountable EVA forcing, which uses the scaling of 1.67 based on a set of sensitivity experiments for the 1815 Tambora eruption. The value used here is derived based on sensitivity tests using the 1991 Pinatubo eruption by comparing the EVA-generated forcing and available CESM forcing (Amann et al. 2003), in order to attain similar atmospheric responses (vertical and surface mean temperatures, surface radiative balances) from both forcings. We have included more detail to explain this in Lines 780-782. Also please consider our response to the comment above about L173.

L741 – are there eruptions that have been excluded or are there none during this time?

- RESPONSE: An eruption occurring in the Southern Hemisphere in 853 CE is excluded in the simulations (Line 785).

L764 – 766 – it would be useful to include this in the main text so that it is clear the volcanic anomalies are not just with reference to 1961-1990

- RESPONSE: change incorporated as suggested: the main text has been updated to include the full detail (now Lines 199-202)

**Technical corrections**

L167 –insert 'the' in front of EVA_H. parameter --> parameters

- REPONSE: change incorporated: included "the model parameters" now in Line 165.

L267 - add 'at 550 nm'

- REPONSE change incorporated as suggested, now Line 268.

L278 – 2019 --> 2020

- REPONSE: change incorporated as suggested, now Line 279.

L286 – add also injection height?

- REPONSE: change incorporated ("injection altitude" is added in to Line 291).

L504 – Northern Hemisphere --> NH

- REPONSE: change incorporated as suggested and all other occurrences have been checked and updated.

L719 – add 'for Phase 4 of PMIP'

- REPONSE: change incorporated as suggested, now Line 756.

L728 – remove 'simulation'

- REPONSE: change incorporated as suggested, now Line 766.

L742 – insert 'A'

- REPONSE: change incorporated (although changed to plural rather than singular): it now states "Mann-Whitney U-tests were used.." in Line 787.

L789 – location --> locations

- REPONSE: change incorporated as suggested, now Line 834.

Appendix F – what do panel a and b refer to? Blue/red labels are inconsistent in caption vs. legend.

- REPSONSE: changes made to incorporate comments: have removed mention of panel a and b (as these are not required here) and have altered the caption to ensure that all is consistent within this figure (now Lines 845-848).

Please check Mount vs. Mt throughout.

- RESPONE: change incorporated: have adopted the common approach of removing 'Mount' or 'Mt' when referring to the eruption after the first use in paper. All other 'Mt' are standardised to 'Mount'.

Additional edits were made to correct and clarify text in Lines 480-500 and 506-523 (historical results) and Lines 654-660 (historical discussion). The overall meaning is unchanged.

Yours sincerely,
Helen Mackay (on behalf of all authors)